# Bioclimatic change as a function of global warming from CMIP6 climate projections

Morgan Sparey[1], Peter Cox[2], and Mark S. Williamson[2,3]

[1]Natural Sciences, Faculty of Environment, Science and Economy, University of Exeter, Exeter, EX4 4QF, UK
[2]Mathematics and Statistics, Faculty of Environment, Science and Economy, University of Exeter, Exeter, EX4 4QF, UK
[3]Global Systems Institute, University of Exeter, Exeter, EX4 4QE, UK

**Correspondence:** Morgan Sparey (ms959@exeter.ac.uk)

**Abstract.** Climate change is predicted to lead to major changes in terrestrial ecosystems. However, substantial differences in climate model projections for given scenarios of greenhouse gas emissions, continue to limit detailed assessment. Here we show, using a traditional Köppen-Geiger bioclimate classification system, that the latest CMIP6 Earth System Models actually agree well on the fraction of the global land-surface that would undergo a major change per degree of global warming. Data from 'historical' and 'ssp585' model runs are used to create bioclimate maps at various degrees of global warming, and to investigate the performance of the multi-model ensemble mean when classifying climate data into discrete categories. Using a streamlined Köppen-Geiger scheme with 13 classifications, global bioclimate classification maps at 2K and 4K of global warming above a 1901 - 1931 reference period are presented. These projections show large shifts in bioclimate distribution, with an almost exclusive change from colder, wetter bioclimates to hotter, drier ones. Historical model run performance is assessed and examined by comparison with the bioclimatic classifications derived from the observed climate over the same time period. The fraction ($f$) of the land experiencing a change in its bioclimatic class as a function of global warming ($\Delta T$) is estimated by combining the results from the individual models. Despite the discrete nature of the bioclimatic classification scheme, we find only a weakly-saturating dependence of this fraction on global warming $f = 1 - e^{-0.14\Delta T}$, which implies about 13% of land experiencing a major change in climate, per 1K increase in global mean temperature between the global warming levels of 1 and 3K. Therefore, we estimate that stabilising the climate at 1.5K rather than 2K of global warming, would save over 7.5 million square kilometres of land from a major bioclimatic change.

## 1 Introduction

Understanding the impacts that climate change will have at a regional level yields vital information for adaptation to climate change. Furthermore, quantifying the performance of climate models is important for the continued improvement of climate models, and for understanding the areas where particular models under perform. There are substantial differences in climate model projections for given scenarios of greenhouse gas emissions (Masson-Delmotte et al., 2021). Climate change is predicted to lead to major changes in terrestrial ecosystems (Pörtner et al., 2022).

Here we use the Köppen-Geiger (KG) bioclimate classification to examine and quantify changes in biome under various levels of projected future global warming within the Coupled Model Intercomparison Project phase 6 (CMIP6) climate models

(Eyring et al., 2016). CMIP6 is an international collaboration to run a standardised set of potential future scenarios with a range of climate models developed at various institutions. The results make a compelling case for the need to further prioritise climate change mitigation policies. However, this may not be immediately clear to public or policy makers. Improved understanding of the consequences of climate change is needed, and climate classification schemes can help in that respect.

The biome of a region is largely dictated by that region's climate. Bioclimates are defined by the preferences of living
organisms. Bioclimate classification empirically separates regions of the globe based on climate data and the geographical distribution of biomes. The KG bioclimate classification scheme is one of the most established, first developed by Wladimir Köppen (Köppen, 1884) and then enhanced by Rudolf Geiger. The original KG classification scheme consists of thirty separate bioclimates based on dominant vegetation as type determined by Köppen's experience as a botanist. These classifications are based on monthly average temperature and precipitation at each location. The seasonality of these variables, combined with
threshold values, determines the bioclimate classification of the region (Peel et al., 2007). Classifications include hot and cold deserts - regions where there is no rainfall, and tropical rainforests - regions where minimum temperature and threshold precipitation is met.

Bioclimate classification systems, such as the KG and Holdridge schemes (Lugo et al., 1999), have been used to map regions and even the entire globe. These maps have been created using observational (Kottek et al., 2006) as well as climate model data,
the latter including CMIP5 (Rahimi et al., 2020; Phillips and Bonfils, 2015) and CMIP6 climate models (Kim and Bae, 2021). Despite the changes and updates suggested by various authors, the classification scheme as originally developed by Köppen, and updated by Geiger is still a highly popular climate classification system. Although bioclimate maps for specific years (such as 2100) have previously been created (Beck et al., 2018), an area that is less explored are global KG climate maps at specific levels of global warming. To remove the leading order uncertainty that arises from different climate model sensitivities
to radiative forcing (Sherwood et al., 2020; Nijsse et al., 2020), and to make our results relevant to the Paris climate targets (Paris-Agreement), here we look at changes in KG classification at different levels of global warming (1.5K, 2K, and 4K). A streamlined KG scheme is also implemented to visually demonstrate the impacts of warming on global biome distribution.

KG classification maps at 1.5K, 2K, and 4K of global warming above reference period levels (taken as the 1901 – 1931 global mean temperature) are presented. Due to the 30 different classifications in the traditional KG scheme, it can be difficult
to identify the changes in bioclimate classification so we present a novel "streamlined" classification system that allows for easy identification of bioclimate change, with a naming scheme that is more intuitive. To quantify this, classification change matrices are also given. By comparing the classifications given by models under the historical experimental run to the known historical observational values, and by assessing model deviation from their initial classifications, we gain insight into the performance and behaviours of individual models as well as the multi-model ensemble mean.

We show there are large shifts in bioclimate distribution under global warming, with an almost exclusive change from colder, wetter bioclimates to hotter, drier ones. Specifically we find the fraction ($f$) of the land experiencing a change in its bioclimatic class has a weakly-saturating dependence on global warming $f = 1 - e^{-0.14\Delta T}$, which implies about 13% of land experiencing a major change in climate, per 1K increase in global mean temperature between the global warming levels of 1 and 3K.

## 2 Methods

 ### 2.1 Köppen-Geiger classification scheme

The Köppen-Geiger (KG) classification scheme has been described extensively in other publications (Peel et al., 2007; Beck et al., 2018). The scheme has also undergone many alterations. Here we follow (Peel et al., 2007), whose criteria for each classification are given in Table 1.

| Classification | | | Criteria | Classification | | | Criteria |
|---|---|---|---|---|---|---|---|
| A | | | $Tmin \geq 18°C$ | D | | | $Tmin \leq 0°C$, $Tmax \geq 10°C$ |
| | F | | $Pmin \geq 6$ cm month$^{-1}$ | | W | | Pswet > 10*Pwdry |
| | S | | $Pmin \geq 100\text{-}(Pyear*10/25)$ | | S | | 3*Psdry <Pwwet, Psdry < 4 |
| | W | | $Pmin <100\text{-}(Pyear*10/25)$ | | F | | Neither W nor S |
| B | | | Pyear*10 <10*Pthresh | | | a | $Tmax \geq 22°C$, Months above $10°C \geq 4$ |
| | W | | Pyear*10 <5*Pthresh | | | b | $Tmax <22°C$, Months above $10°C \geq 4$ |
| | S | | $Pyear*10 \geq 5*Pthresh$ | | | c | 0 <Months above 10°C <4, not A or B or D |
| | | h | $Tavg \geq 18°C$ | | | d | Tmin <-38°C, 0 <Months above 10°C <4 |
| | | k | $Tavg <18°C$ | E | | | Tmax <10°C |
| C | | | $0°C < Tmin <18°C$, $Tmax \geq 10°C$ | | T | | $0°C < Tmax <10°C$ |
| | W | | Pwdry <Pswet/10 | | F | | $0°C \geq Tmax$ |
| | S | | Pwwet > 3*Psdry, Psdry <4 | | | | |
| | F | | Neither W nor S | | | | |
| | | a | $Tmax \geq 22°C$, Months above $10°C \geq 4$ | | | | |
| | | b | $Tmax <22°C$, Months above $10°C \geq 4$ | | | | |
| | | c | 0 <Months above 10°C <4, not A or B | | | | |

**Table 1.** Classification criteria for the Köppen-Geiger classification scheme. Tmin = Average temperature of month with lowest average temperature. Tmax = Average temperature of month with highest average temperature. Pmin = Average precipitation of driest month. Pmax = Average precipitation of wettest month. Tavg = Mean annual temperature. Pyear = Mean annual precipitation. Pthresh varies according to the following rules (if 70% of Pyear occurs in winter then Pthresh = 2 x Tavg, if 70% of Pyear occurs in summer then Pthresh = 2 x Tavg + 28, otherwise Pthresh = 2 x Tavg + 14). , Psdry = precipitation of the driest month in summer, Pwdry = precipitation of the driest month in winter, Pswet = precipitation of the wettest month in summer, Pwwet = precipitation of the wettest month in winter. In the northern Hemisphere Summer is defined as AMJJAS and Winter as ONDJFM, the opposite is true for the Southern hemisphere. Due to overlapping criteria, dry (B) climates are prioritised above all others. Temperature is in °C and precipitation is cm month$^{-1}$ and cm year$^{-1}$. Here we follow (Peel et al., 2007).

These classifications have three differences to those described by (Köppen, 1936). First, C and D climates follow a 0°C

 threshold instead of -3°C (Russell, 1931). Secondly, BW and BS are distinguished using a 70% threshold for precipitation

seasonality (Peel et al., 2007). Finally, climates C and D subclassifications s and w are made mutually exclusive (Peel et al., 2007). In this analysis, each month is set to have the same length of time – one twelfth of a year.

The KG system has been applied to a broad spectrum of scientific interests, including to locally adjust an irradiation model (Every et al., 2020), in hydrological studies (Peel et al., 2001), and in modelling the distribution of Lyme disease (Cox et al., 2021).

The outcome of competition between different plant types varies depending on the climatic conditions, such that the long-term equilibrium biome (which is a mixture of different plant types) will also vary with the climate. This is the underlying basis for bioclimatic schemes such as Koppen-Geiger, which is used here to classify the climate rather than to predict biome shifts. Changes in the actual distribution of vegetation also depend on the direct effects of changes in carbon dioxide and nutrient availability, and may take decades to materialise (e.g. because the rate of climate change is significant compared to the characteristic multi-decadal timescales of a forest). These changes are best predicted with complex Dynamical Global Vegetation Models (DGVMs), which are based-on detailed representations of plant physiology and demographic processes (Argles et al., 2022). Although bioclimatic schemes are no substitute for DGVMs to predict vegetation changes, they have the advantage of being transparently simple and offering a more intuitive demonstration of the nature of a projected climate change. It is in this spirit that the Koppen-Geiger bioclimatic scheme is applied in this study.

### 2.1.1 Streamlined Köppen-Geiger classification scheme

A key goal of bioclimatic classifications is to illustrate climate change in a way that is intuitive. To this end we designed a simplified Köppen-Geiger scheme which combines classifications to make changes clearer in both scale and the nature of projected transitions. Additionally, the new scheme implements a more traditional naming system. A breakdown of this streamlined system, and the constituent traditional classifications involved in each of the thirteen streamlined classifications is given in Table 2.

Difference maps are also plotted to demonstrate the geographical locations of major transitions between bioclimatic classifications. These difference maps plot the ten largest transitions globally (by total land area). Areas for which less than 66% of the models agree are hatched, demonstrating that in these regions the results are less certain.

Classification change matrices are used to quantify bioclimate transitions in terms of global land area, at key levels of global warming. The columns represent the initial classification coverage, and the rows indicate the altered classification distribution. Shading highlights the size of changes, in terms of the projected change as a fraction of the initial area of a given bioclimatic class.

### 2.2 Climate Model and Observational Data

Historical observations of monthly mean temperature and precipitation are from the CRU TS v. 4.05 dataset (Harris et al., 2020). Analogous climate model data comes from the 'historical' CMIP6 experiments (Eyring et al., 2016). Models within the CMIP6 multi-model ensemble were chosen which had readily available historical experiment data, and achieved a minimum of 4K warming under the ssp585 scenario. These models are listed in Table 3.

| Streamlined Classification | Traditional Classifications |
| --- | --- |
| Desert | BWh, BWk |
| Semi-Arid | BSh, BSk |
| Tropical Rainforest | AF |
| Tropical Monsoon | AM |
| Tropical Savanna | AW |
| Mediterranean | CSa, CSb, CSc |
| Subtropical | CWa, CWb, CWc, CFa |
| Oceanic | CFb, CFc |
| Continental hot-summer | DFa, DSa, DWa |
| Continental cold-summer | DFb, DSb, DWb |
| Sub Arctic | DFc, DFd, DSc, DSd, DWc, DWd |
| Arctic Tundra | ET |
| Icecap | EF |

**Table 2.** Breakdown of the streamlined classification scheme and the assignment of traditional classifications within the new scheme.

CMIP6 model data was regridded to 0.5˚ by 0.5˚, the same spatial resolution as CRU TS observations. Antarctica was
excluded as observations are limited in this region, and we do not expect substantial changes in bioclimatic classification in
this region.

The model output data is typically at a coarser resolution than the underlying 0.5˚ climatology. The anomaly corrected fields
therefore contain spatial variability that is solely due to the underlying climatology at scales which are not resolved by a model.
This also implies that the diagnosed changes in bioclimatic types (which are dependent on the model anomalies) tend to be
somewhat smoother at these finer spatial scales.

### 2.3    Model performance assessment

Comparison of KG observed classifications with the CMIP6 models simulated classifications is made for the years 1901-2014.
To reduce the effect of short-term variability model and observational data are smoothed with a 30 year centred rolling mean.

The ability of individual models in the CMIP6 ensemble to simulate KG classifications of observational data during the
historical period is assessed in two ways: (i) Percentage land area that a model has correctly classified for each year rela-
tive to observations. (ii) Percentage change in land area classification at each year compared to the initial mean 1901-1931
classifications.

| Model | Institution | Frequency | Nominal Resolution | Publication |
|-------|-------------|-----------|--------------------|-------------|
| CanESM5 | CCCma | mon | 100 km | (Swart et al., 2019d) (Swart et al., 2019a) |
| CanESM5-CanOE | CCCma | mon | 100 km | (Swart et al., 2019c) (Swart et al., 2019b) |
| CESM2 | NCAR | mon | 100 km | (Danabasoglu, 2019b) (Danabasoglu, 2019a) |
| CESM2-WACCM | NCAR | mon | 100 km | (Danabasoglu, 2019c) (Danabasoglu, 2019d) |
| IPSL-CM6A-LR | IPSL | mon | 100 km | (Boucher et al., 2018) (Boucher et al., 2019) |
| UKESM1-0-LL | Met Office Hadley Centre | mon | 100 km | (Tang et al., 2019) (Good et al., 2019) |
| ACCESS-CM2 | CSIRO-ARCCSS | mon | 250 km | (Dix et al., 2019b) (Dix et al., 2019a) |
| AWI-CM-1-1-MR | NCAR | mon | 100 km | (Danek et al., 2020) (Semmler et al., 2019) |
| CAS-ESM2-0 | UCI | mon | 100 km | (Chai, 2020) (Cas, 2018) |
| EC-Earth3 | EC-Earth-Consortium | mon | 100 km | (EC-Earth-Consortium, 2019b) (EC-Earth-Consortium, 2019a) |
| EC-Earth3-Veg | EC-Earth-Consortium | mon | 100 km | (EC-Earth-Consortium, 2019d) (EC-Earth-Consortium, 2019c) |
| TaiESM1 | AS-RCEC | mon | 100 km | (Lee and Liang, 2020b) (Lee and Liang, 2020a) |

**Table 3.** Details of the models used in this study.

## 2.4 Maps of KG classification versus global warming

Future KG classification maps under 1.5, 2 and 4K of annual mean global warming above reference period levels were created

from the CMIP6 'ssp585' 2015-2100 future scenario. We used ssp585 because all models pass 4K under ssp585, which enables us to define changes in bioclimatic zones consistently for these different levels of global warming.

The timing of each warming level is found from the centred 30 year annual mean global surface air temperature above the model's reference temperature, here defined as 1901 – 1931. Monthly mean anomalies of precipitation and surface air temperature are calculated relative to this same reference period. Model outputs are anomaly corrected to agree with the

120 observational over the period 1901-1931. This is done by calculating anomalies relative to that period for each model and then

adding these anomalies to the observational climatology. Multi-model ensemble mean KG classification maps are calculated using the multi-model ensemble mean of the anomalous temperature and precipitation fields at each warming level. As usual in climate modelling, we focus primarily on the ensemble mean, although we note that ensemble mean and ensemble median climates have been found to be very similar (Sillmann et al., 2013; Kim et al., 2020; Li et al., 2021). These maps are hatched to indicate where models disagree on the classification. A grid point is hatched if less than 66% of the models agree on the classification at the point.

## 3 Results and discussion

### 3.1 Model performance assessment

To gain insight into the behaviour of individual models, we create KG maps of individual models and compare these with maps derived from the observed climate. As expected, there is variation in the classification distribution of models and the observational data. For example, desertification in the Amazon is apparent in CanESM5 and CanESM5-CanOE models (Appendix A). This may show that these models have a tendency towards reduced precipitation in the tropics when compared to other models. Another area of disagreement between the models is the change of biome classification in northern Eurasia and America at various levels of global warming. The multi-model ensemble mean model state however reduces the effect of individual model discrepancies and compares favourably with observations.

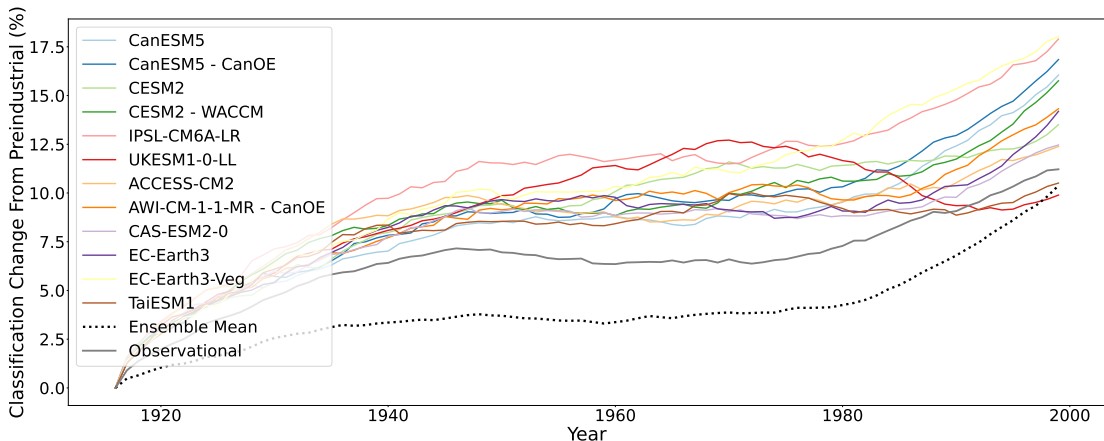

**Figure 1.** The percentage land area change in K-G classification for each model versus year through the 20th century (without anomaly correction). The equivalent trajectory based on the observed climate is shown for comparison in dark-grey. The dotted line shows the trajectory derived from the ensemble mean climate .

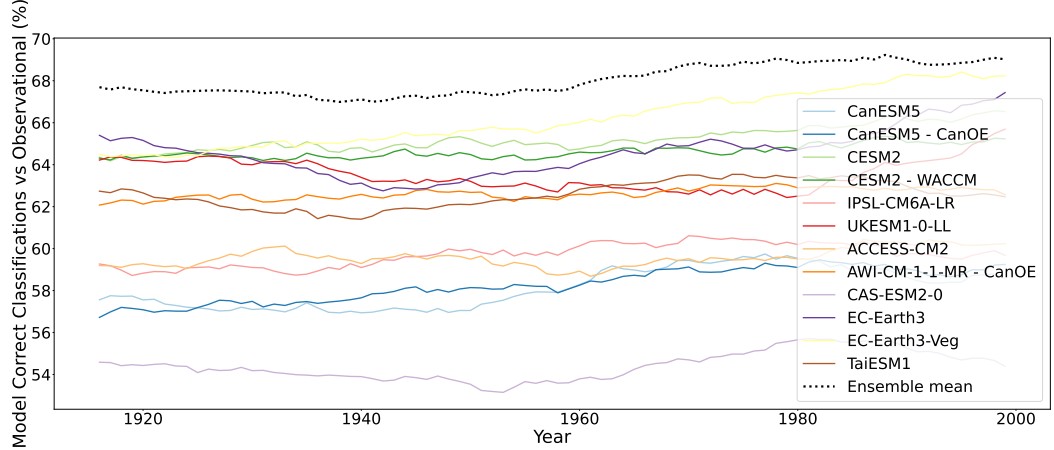

**Figure 2.** The percentage land area correctly classified (without anomaly correction).

In Figure 1 simulated classification changes from the CMIP6 historical runs are compared to those calculated from the observed climate. The CMIP6 models broadly capture the degree of expected global classification change. All models show a similar behaviour – a large change in classifications at the start of the observed period until 1940, the mid-century then presents an approximately constant set of classification with very little change until 1980, where again all models display further changes
in climate classification. Although the multi-model ensemble mean follows the same pattern as the individual models and the

observational data, it shows a lesser degree of change throughout the observed time period. This reduced variation is inherent to the nature of this ensemble mean; large changes in individual models have their impact reduced in the meaning process. This may lead to the multi-model ensemble mean displaying a similar but mitigated and delayed trend 'lagging' the individual models and the observational when creating discrete classes from climate data.

To assess the performance of individual models and their multi-model ensemble mean in classifying the bioclimate distribution according to observation-based KG for a particular year, the percentage land area correctly classified by each model every year according to observation-based KG is shown in Figure 2. The results show that the ensemble mean is one of the best performing for classification. This is in contrast to Figure 1 which showed the ensemble mean was one of the worst performing for classification change. The reason is also likely due to the reduced variation in data resulting from the averaging process in the creation of the multi-model ensemble mean dataset. The impact of 'extreme' values present in each model are averaged out in the multi-model ensemble mean provided they are distributed around the 'true' climate values. This would suggest that for individual time points, the ensemble mean is likely to provide the the most reliable projection. The results from Figure 1 and Figure 2 give insight into the behaviour of ensemble mean datasets and when their application is appropriate. Traditionally the ensemble mean has been taken as the most likely scenario and therefore the most representative of the real-world climate. The results presented here indicate that although the ensemble mean is appropriate for assessing model output at individual points, the ensemble mean does not accurately display the variability evident in observed climate data.

## 3.2 Maps of KG classification versus global warming

Figure 3 shows the multi-model ensemble model mean KG classification for 1.5K, 2K and 4K of global warming above the reference period, as well as the no warming classifications. Plots for individual models for the reference period without anomaly correction, and at 1K, 1.5K, 2K, 3K, and 4K of global warming with anomaly correction, are shown in appendix A under the traditional scheme, and Appendix B under the streamlined scheme. Comparison to the reference climate suggests that there could be dramatic changes in bioclimate classification, particularly in the mid- to high latitudes, as the planet warms. There is less agreement in classification at the boundaries of classified regions, this is expected as the models will likely be split in classifying grid cells as one of two classifications. These changes become more apparent in Figure 4, which use the streamlined KG classification scheme and highlights the ten largest bioclimatic shifts for each level of warming. Figure 4 again demonstrates less certainty at the classification change boundaries, however the degree of agreement between the models, even at 4K of warming, is surprisingly high.

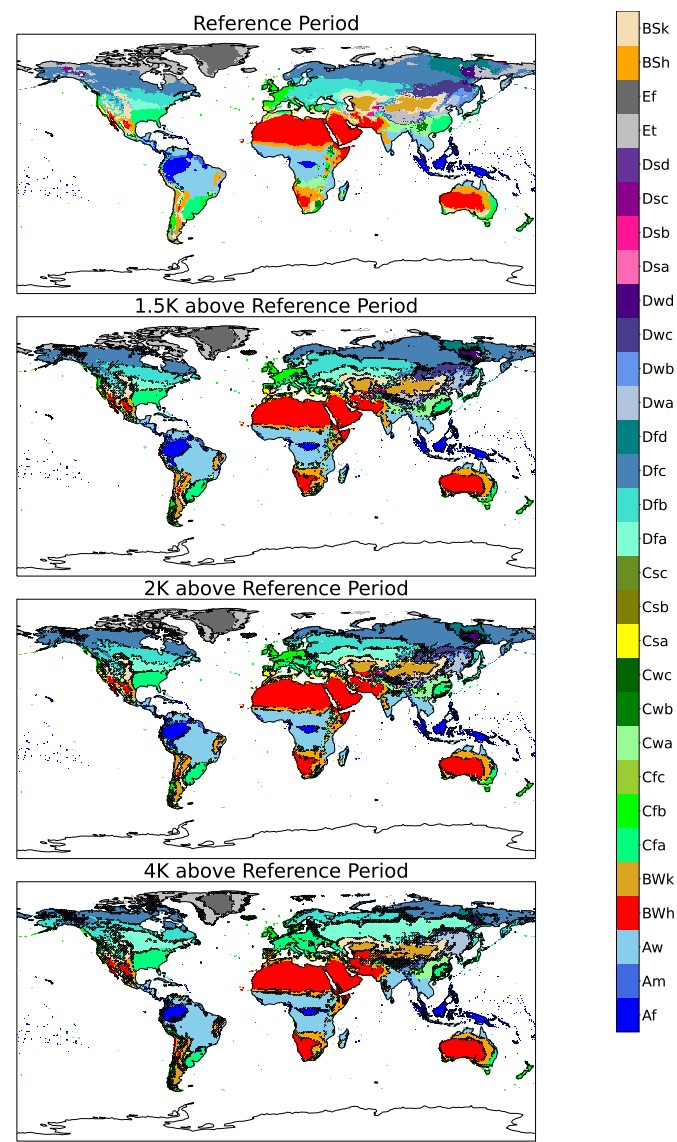

**Figure 3.** Maps of the original K-G classifications calculated from the multi-model ensemble mean for the reference period and 1.5K, 2.0K, 4.0K of global warming relative to the reference period. These were calculated from the SSP585 runs by anomaly correcting relative to the observed reference climate. Hatching is present when less than 66% of models agree on a region's classification.

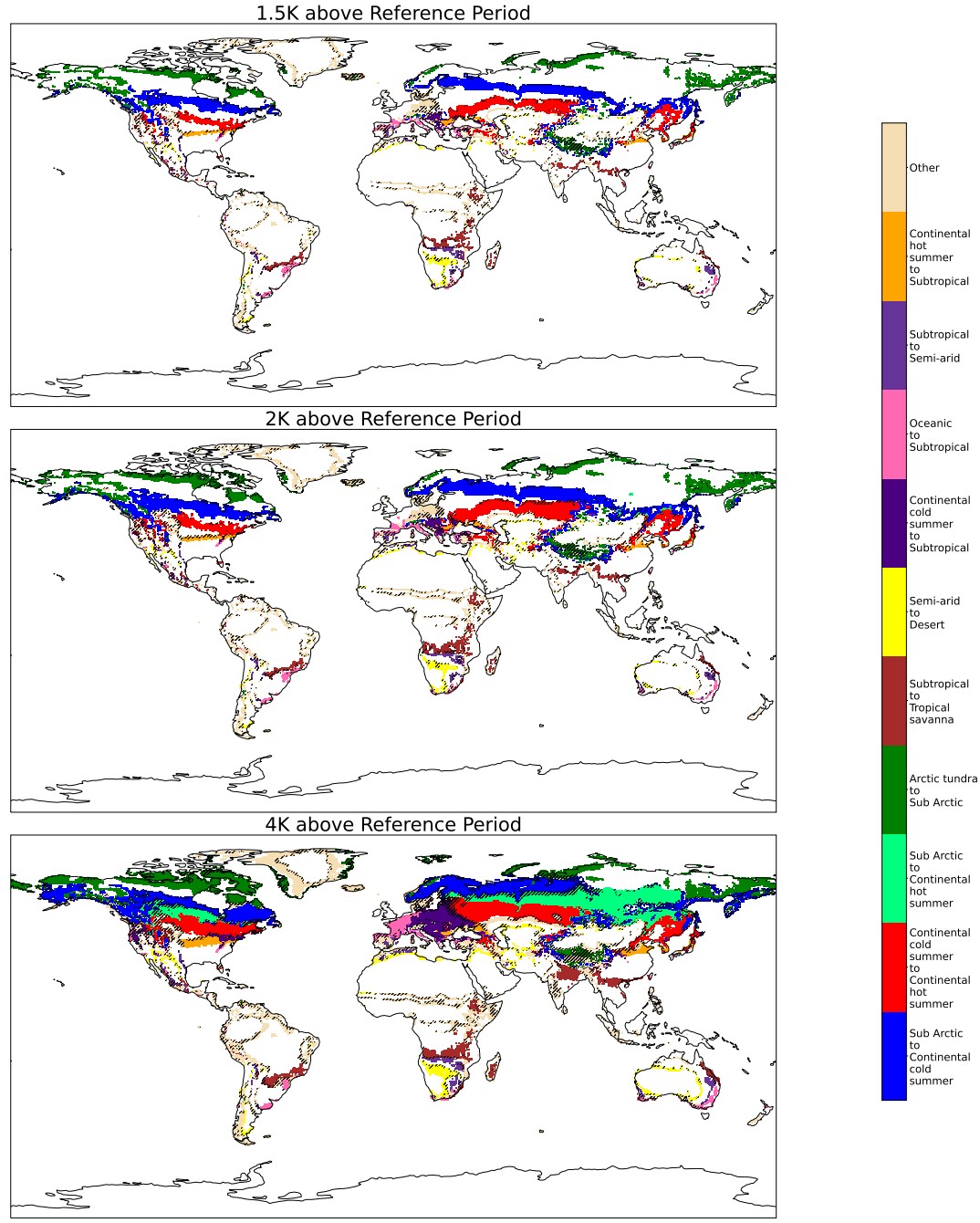

**Figure 4.** Multi-model ensemble mean difference maps highlighting the ten largest classification changes for 4K of warming above the reference period using the streamlined classification system. Hatching is present when less than 66% of models agree on a region's classification change.

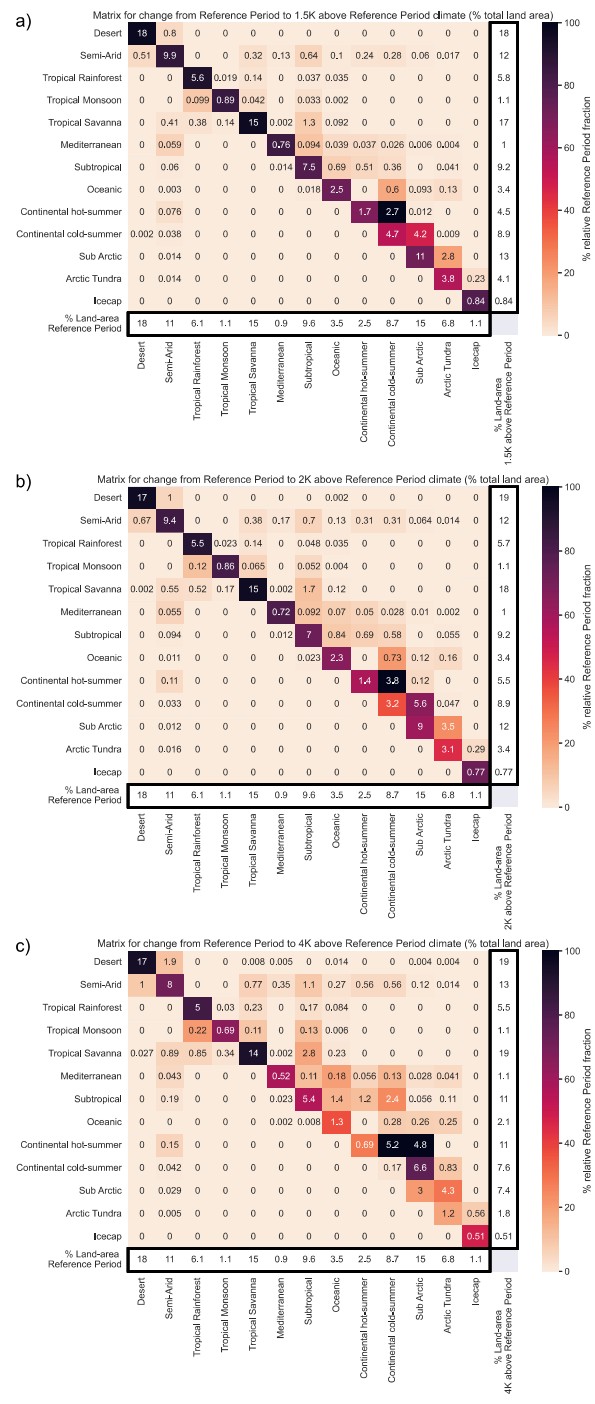

**Figure 5.** Multi-model ensemble mean difference matrices highlighting the classification changes for levels of warming above the reference period of a) 1.5K, b) 2K, c) 4K, using the streamlined classification system.

These shifts are almost exclusively from wetter and colder classes to drier and hotter ones as the global temperature increases. This agrees strongly with the results found by Feng et al. (2014), which under CMIP5's RCP8.5 scenario, suggested bioclimatic

shifts toward warmer and drier types across the global region with climate change. Large areas undergo desertification in the southern hemisphere. The majority of North America and Northern Eurasia has a shift towards warmer climates as Sub-arctic gives way to continental cold summer, and continental cold summer is replaced by continental warm summer. All changes in classification with the streamlined KG scheme are quantified in Figure 5. Although the KG scheme exclusively maps climate, the biologic implications of these changes can be seen. The Sub-arctic region has historically been dominated by the boreal

forest (Kayes and Mallik, 2020) though there is evidence suggesting a slow northward migration of non-native warm-adapted tree species into historically boreal areas (Viacheslav et al., 2007; Boisvert-Marsh et al., 2014). However, the rate of global warming far exceeds the rate of boreal migration due to limits on the tree's ability to migrate (McKenney et al., 2007). With the continued reduction of the sub-arctic bioclimate we predict a likely continuation of these trends. The indicated shift in the Amazon from tropical rainforest to savanna can also be seen, an expansion of white-sand savannas in the Amazon has already

been found (Flo, 2021).

At 4K areas of classification change represent over 33% of land area. The change in % of total land-area in Figure 5c gives some alarming perspectives, for example, at +4K Arctic Tundra is indicated to cover over a quarter less land-area than in in the reference period. At 2K the models already project substantial changes to the global distribution of bioclimates; at 4K these changes become even more pronounced.

In Table 4 we give the percentage change in global land area of each of the streamlined classifications per degree of warming assuming the dependence is linear up to 4K of warming. Linear dependence is a good approximation for most classifications with all but three having $r^2 > 0.9$. The three poorly fitting classifications, those for Mediterranean, subtropical, and continental cold-summer bioclimates, may be transitory classifications whose peak or minimum land area coverage is within the 4K range the linear equation is based on. Classifications predicted to decrease in global fraction under global warming (with good $r^2$)

are tropical rainforest, oceanic, sub Arctic, Arctic tundra and icecap, the largest decrease of which globally is sub Arctic $(2.03 \, \% K^{-1})$. Classifications that increase under global warming with good certainty are desert, semi arid, tropical monsoon, tropical savanna and continental hot-summer with the largest increase predicted to be continental hot-summer $(2.18 \, \% K^{-1})$. Raw plots for these fits without lines fitted can be found in Appendix C2.

### 3.3   Sensitivity of bioclimate to global warming

Figure 6 displays a weakly-saturating increase with global warming, the fraction of land area that experiences a change in classification follows Eq. (1):

$$f = 1 - e^{-k\Delta T} \tag{1}$$

where $f$ is the fraction of land area that experiences a change in bioclimatic classification, $\Delta T$ is the global warming relative to the reference period climate, and $k$ is a fitting parameter. The mean response across the models suggests a value of $k \sim 0.14$

$K^{-1}$. This was calculated to have a coefficient of determination of 0.84. For the range of global warming of particular interest

| Classification | Change in global land area coverage (%) per degree of warming ($K$) | Coefficient of determination ($r^2$) |
|---|---|---|
| Desert | $0.30\,\%K^{-1}$ | 0.93 |
| Semi Arid | $0.35\,\%K^{-1}$ | 0.94 |
| Tropical Rainforest | $-0.26\,\%K^{-1}$ | 0.96 |
| Tropical Monsoon | $0.09\,\%K^{-1}$ | 0.97 |
| Tropical Savanna | $1.01\,\%K^{-1}$ | 0.97 |
| Mediterranean | $0.07\,\%K^{-1}$ | 0.33 |
| Subtropical | $0.25\,\%K^{-1}$ | 0.20 |
| Oceanic | $-0.35\,\%K^{-1}$ | 0.94 |
| Continental Hot-Summer | $2.18\,\%K^{-1}$ | 0.98 |
| Continental Cold-Summer | $-0.25\,\%K^{-1}$ | 0.21 |
| Sub Arctic | $-2.03\,\%K^{-1}$ | 0.97 |
| Arctic Tundra | $-1.24\,\%K^{-1}$ | 0.95 |
| Icecap | $-0.13\,\%K^{-1}$ | 0.99 |
| Streamlined Total Change | $10.85\,\%K^{-1}$ | 0.99 |
| Traditional Total Change | $11.91\,\%K^{-1}$ | 0.97 |

**Table 4.** Percentage change in the global land area of each of the streamlined classifications per degree of global warming. Change is based on linear approximations of results from reference period to 4K of warming

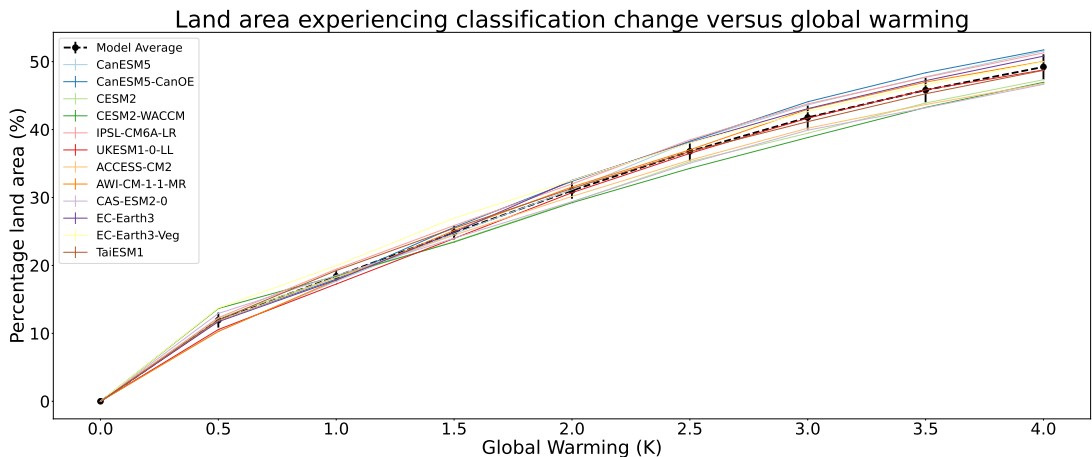

**Figure 6.** The percentage of land area projected to see a change in bioclimate as a function of global warming, using the traditional Köppen-Geiger classification with anomaly correction. Note the robust agreement between models, which implies a multi-model ensemble mean change which is well approximated by: $f = 1 - e^{-0.14\Delta T}$.

to the Paris climate agreement (1 to 3 degrees of warming) the land area experiencing a change in bioclimatic classification is approximately 13% of the global land per Kelvin of global warming. The total land area (neglecting Antarctica) is approximately 146 million square kilometres, so this implies a bioclimatic change for 18.9 million square kilometres of land per degree of warming between 1K and 3K. This highlights the benefits of keeping global warming to 1.5K as opposed to 2K of warming, as the 0.5K difference represents an additional bioclimatic change for over 7.5 million square kilometres of land. These bioclimatic changes being increases of 1% in land area of desert, tropical savanna and continental hot summer biomes. Sub-Arctic and Arctic tundra decrease by 1 and 0.7% respectively. Between 1.5K and 2K there are relatively small or no changes between the remaining classifications in figure 5a and figure 5b.

Previous studies of classification change with global warming have been regional, studies such as Kim and Bae (2021) suggest a classification area change of approximately 15% of Asian monsoon regions at 2K of warming, regional assessments at the equator or in the southern hemisphere are likely to under represent global changes in classification however as the majority of classification change is predicted to be north of 30°N (Feng et al., 2014). A quantitative distribution of climate classification changes between global warming levels of 1.5K and 2K can be seen in Appendix C (note that this breakdown uses the streamlined KG system and subsequently will not represent all changes included in Figure 6).

## 4   Conclusions

Despite the difference in climate projections for given greenhouse gas emissions, we present strong evidence that climate models agree well on the extent of bioclimatic change the global land-surface will undergo per degree of global warming. The Köppen-Geiger scheme has been used to present the impact of global warming at 1.5K, 2K, and 4K of warming above reference period levels in the form of climate maps – showing the global distribution of bioclimates, and as graphs and classification change matrices, at various levels of warming.

Bioclimate classifications are fundamentally climate classifiers, but they are designed to represent and correlate with biome distribution. In this way the warming maps and classification changes represent tangible shifts in the global distribution of ecosystems, giving insight into the nature of Earth at various levels of warming. This paper also uses the Köppen-Geiger scheme as a method for climate model verification which is relevant to the impacts of climate change on ecosystems. The Köppen-Geiger maps at levels of global warming demonstrate the impact that climate change could have. The transition matrices present an easily interpretable method for understanding and quantifying the scale of all classification changes. The results presented by the maps and matrices predict large changes in global bioclimate distribution, with hotter, drier bioclimates expanding and colder, wetter bioclimates shrinking and moving further towards the poles.

The combination of the techniques presented in this paper indicate that the impact of global warming on KG bioclimates is roughly linear for levels of warming between 1 and 3K. We find that 13% of land could experience a substantial change in bioclimate per $^o$C of global warming.

*Data availability.* Based on publicly available CMIP6 data

# Appendix A

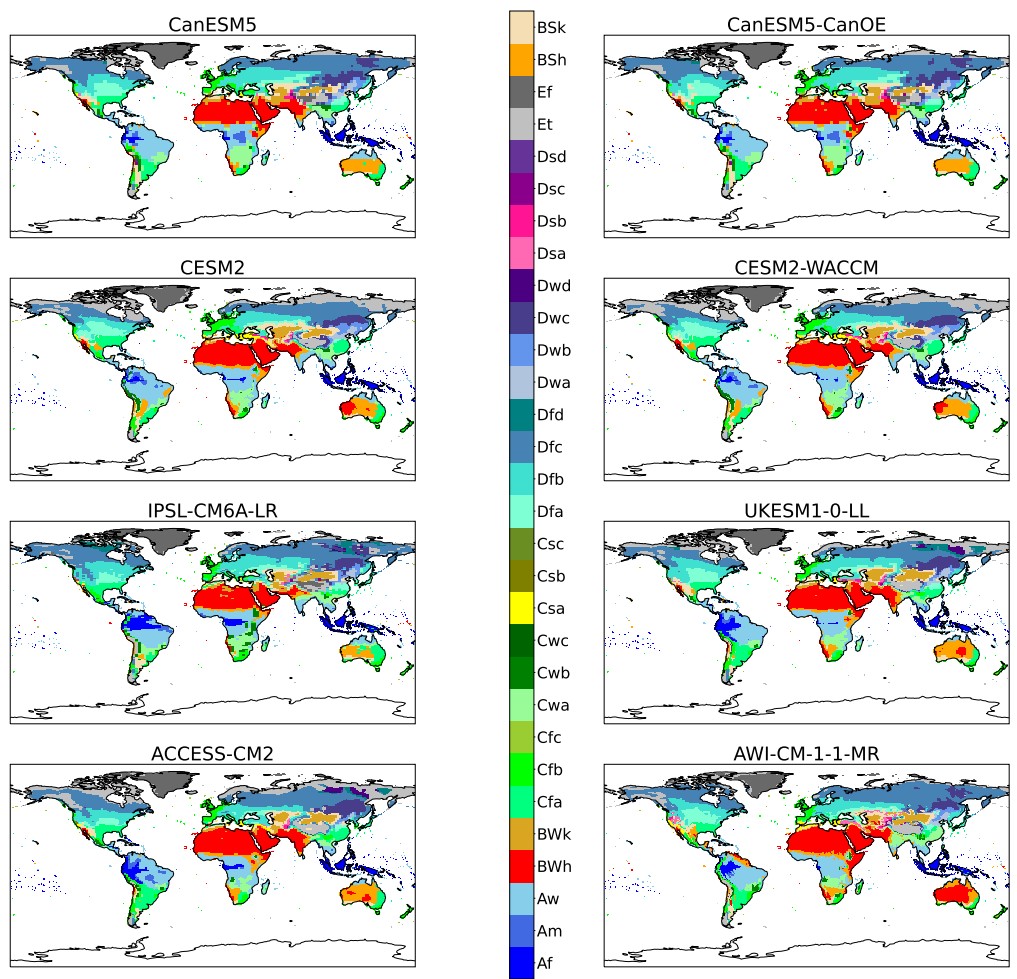

**Figure A1.** Maps of KG classifications for each model for the reference period (1901 - 1931), without anomaly correction.

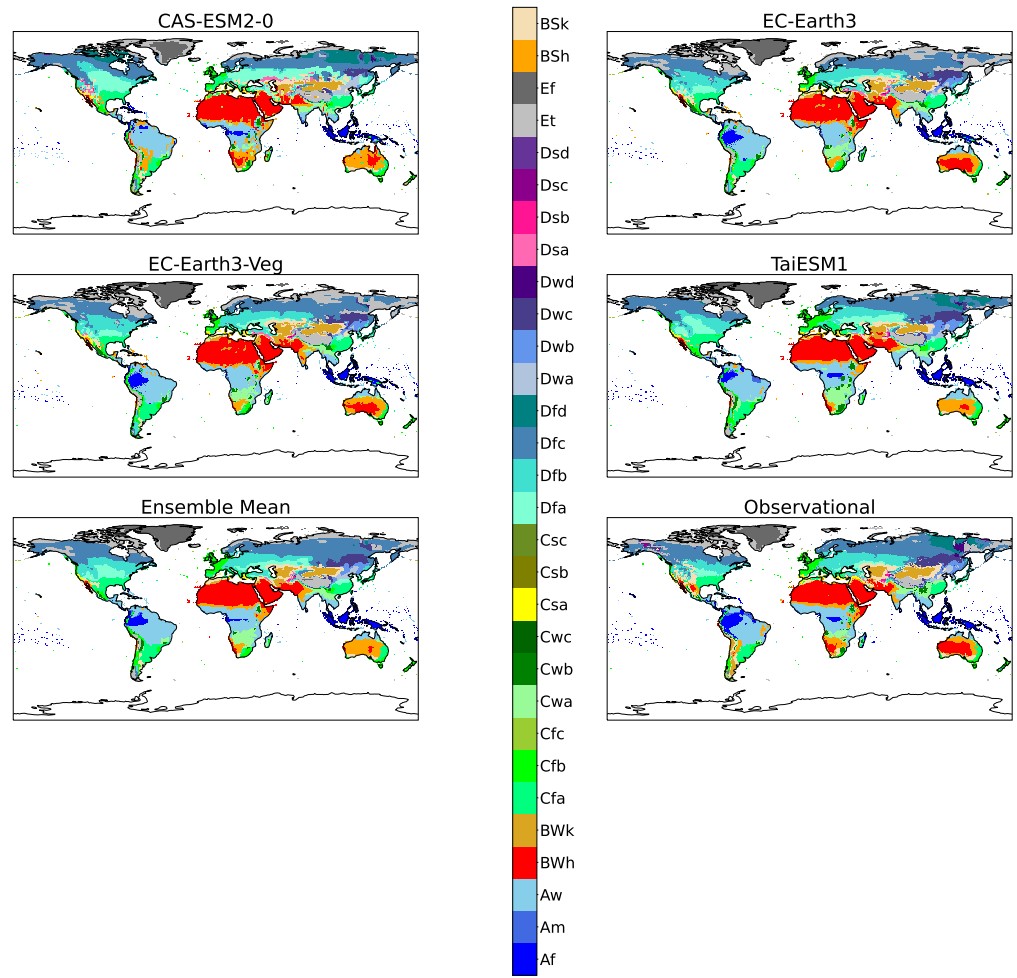

**Figure A1. Continued.**

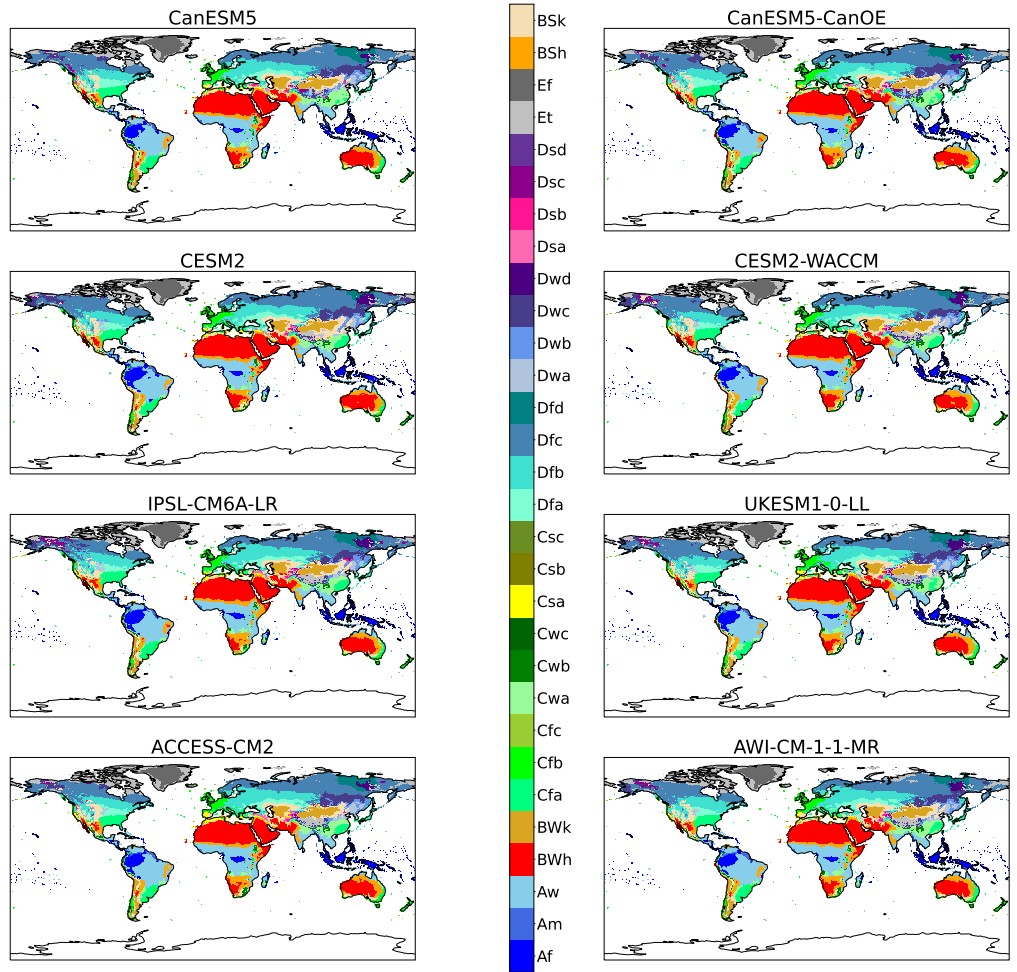

**Figure A2.** Maps of KG classifications for each model at +1K, with anomaly correction.

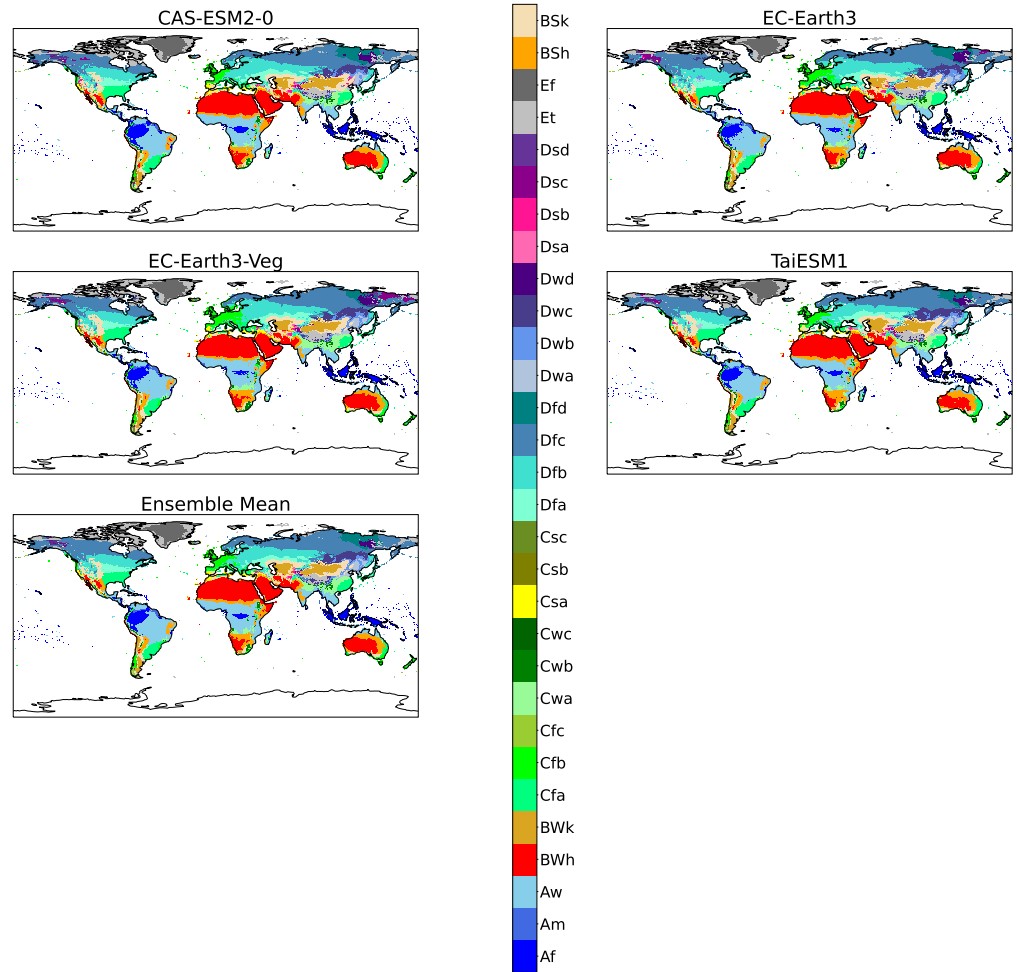

**Figure A2. Continued.**

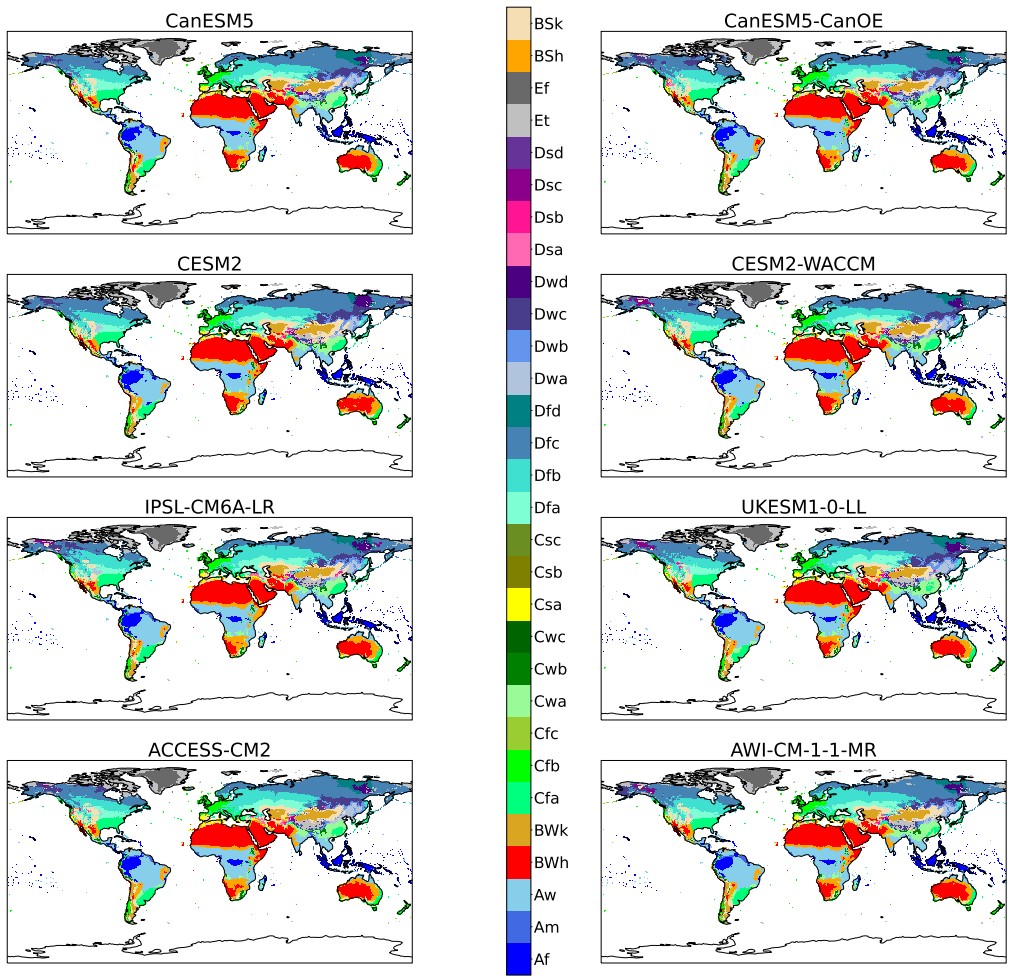

**Figure A3.** Maps of KG classifications for each model at +1.5K, with anomaly correction.

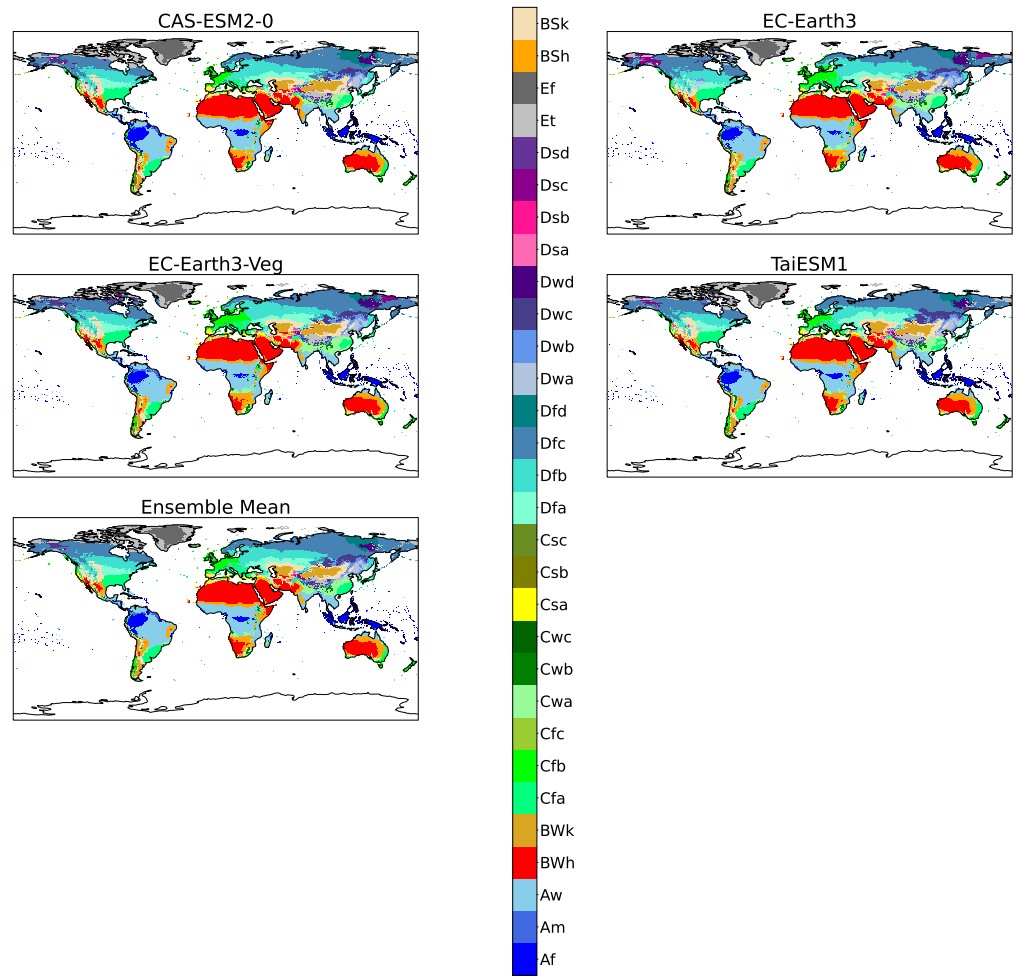

**Figure A3. Continued.**

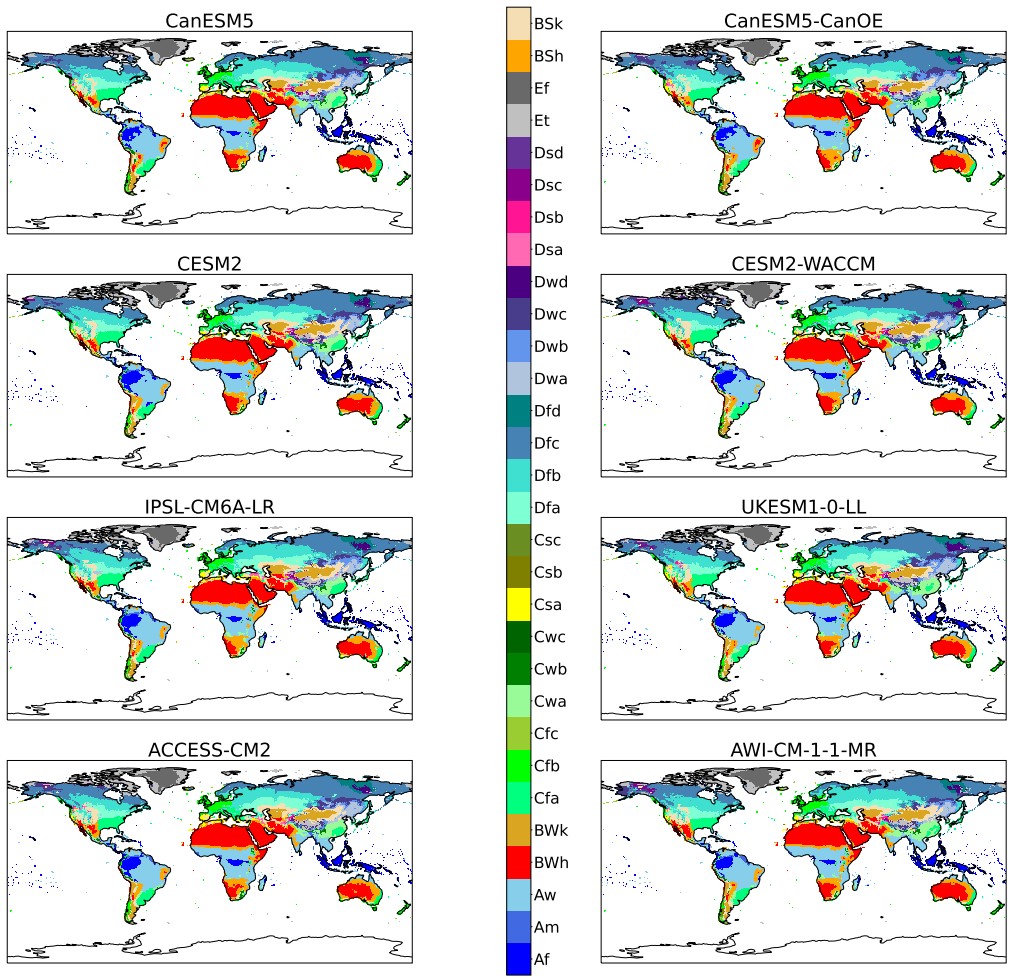

**Figure A4.** Maps of KG classifications for each model at +2K, with anomaly correction.

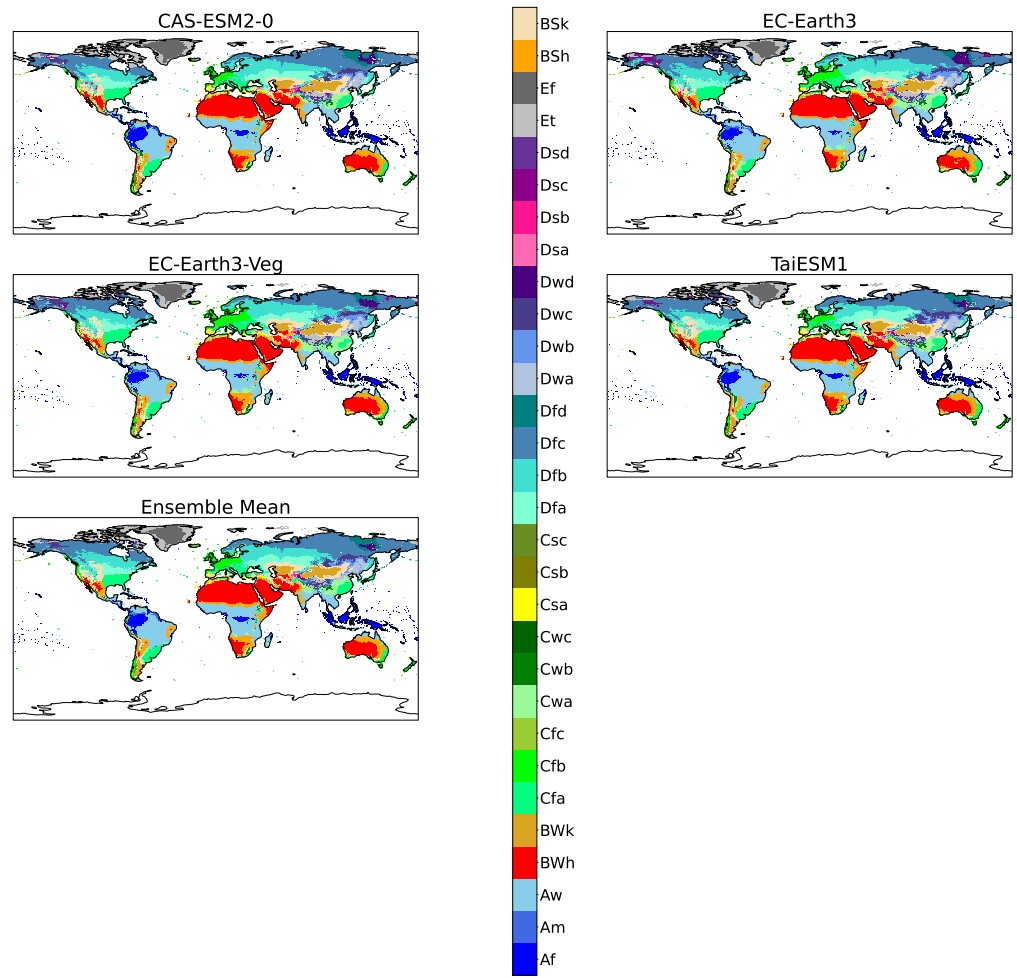

Figure A4. Continued.

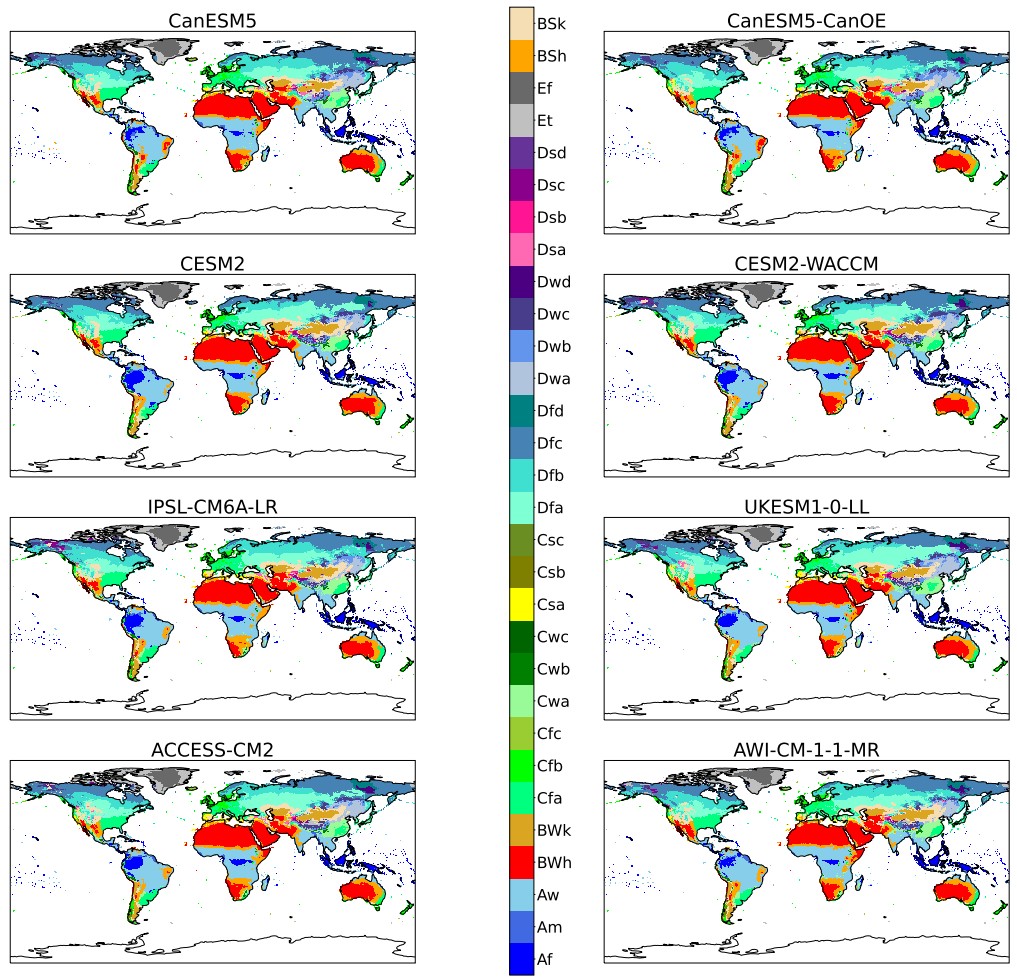

**Figure A5.** Maps of KG classifications for each model at +3K, with anomaly correction.

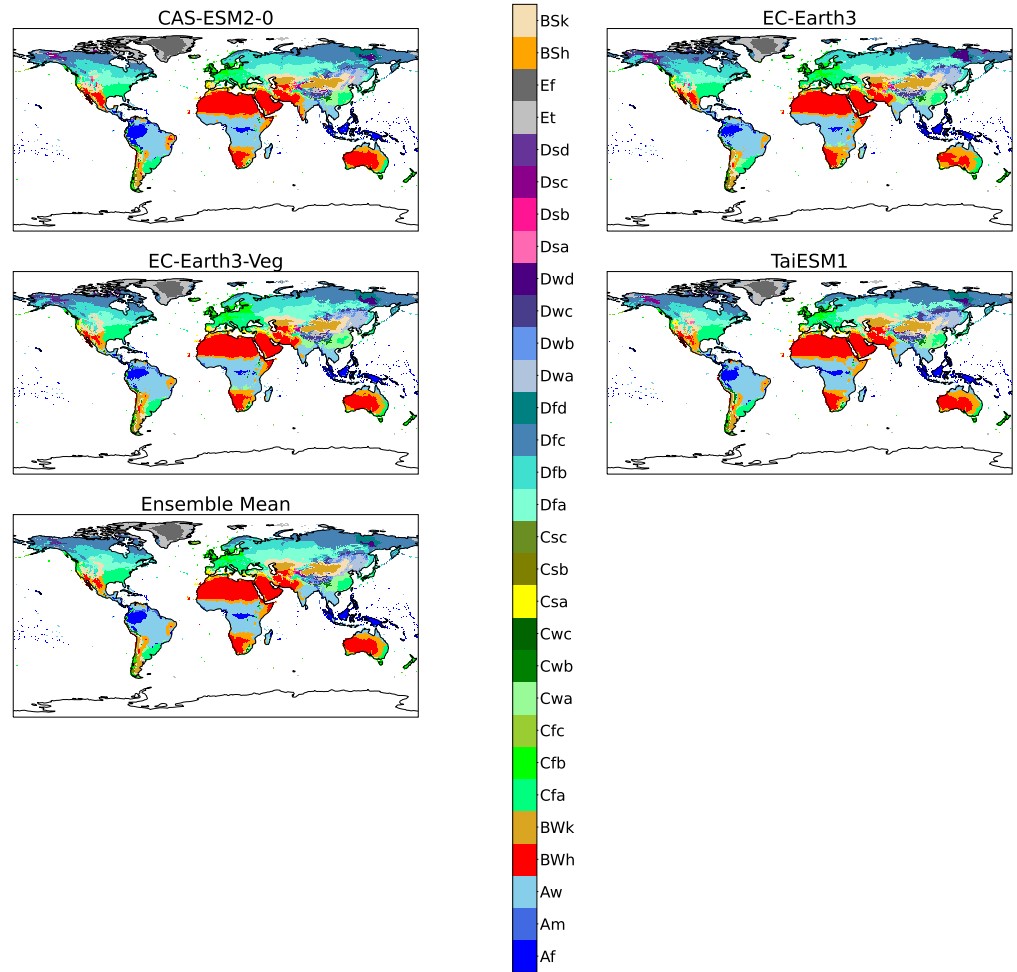

**Figure A5. Continued.**

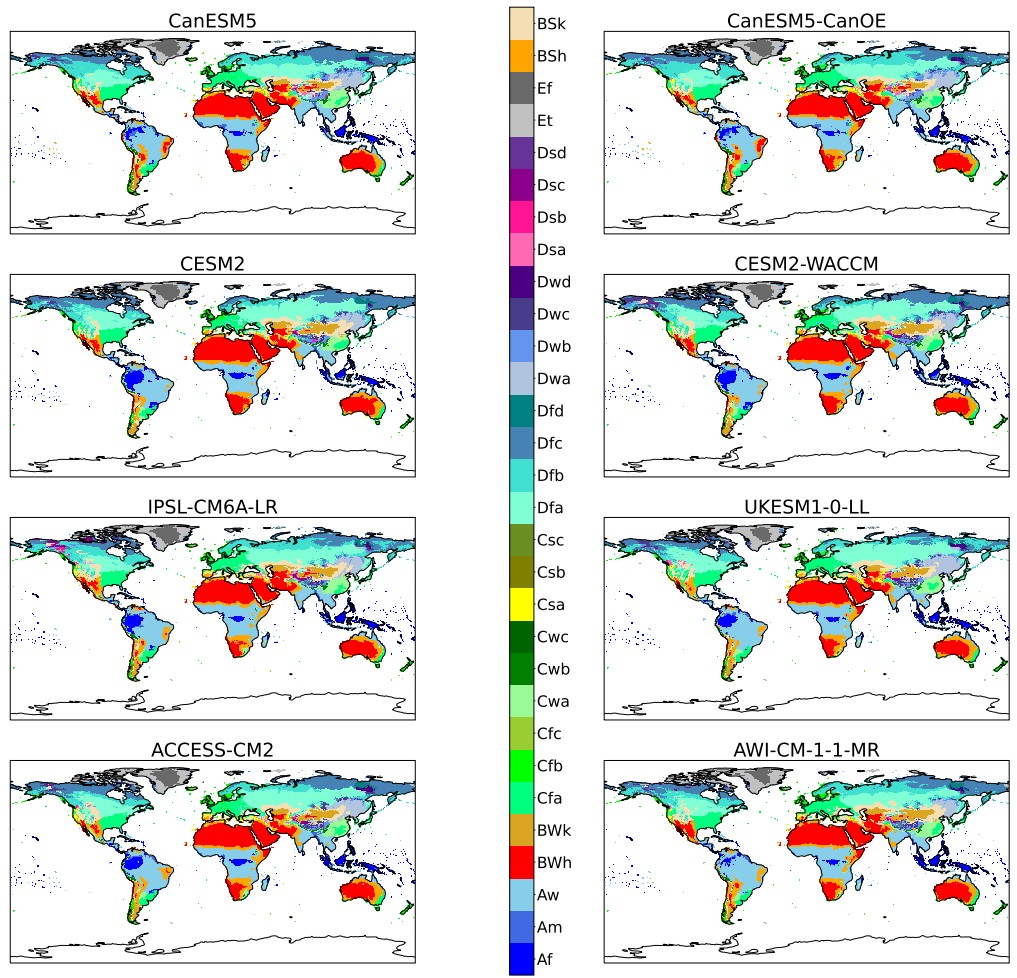

**Figure A6.** Maps of KG classifications for each model at +4K, with anomaly correction.

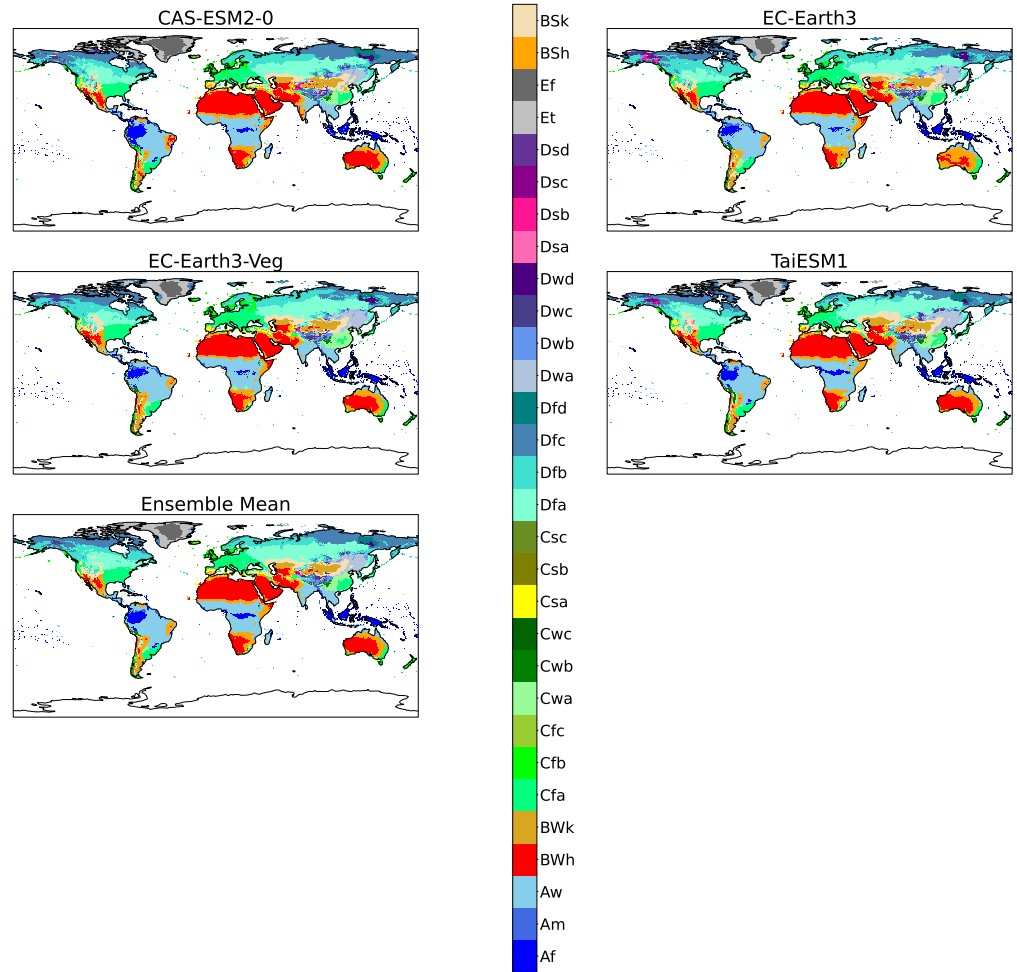

**Figure A6. Continued.**

# Appendix B

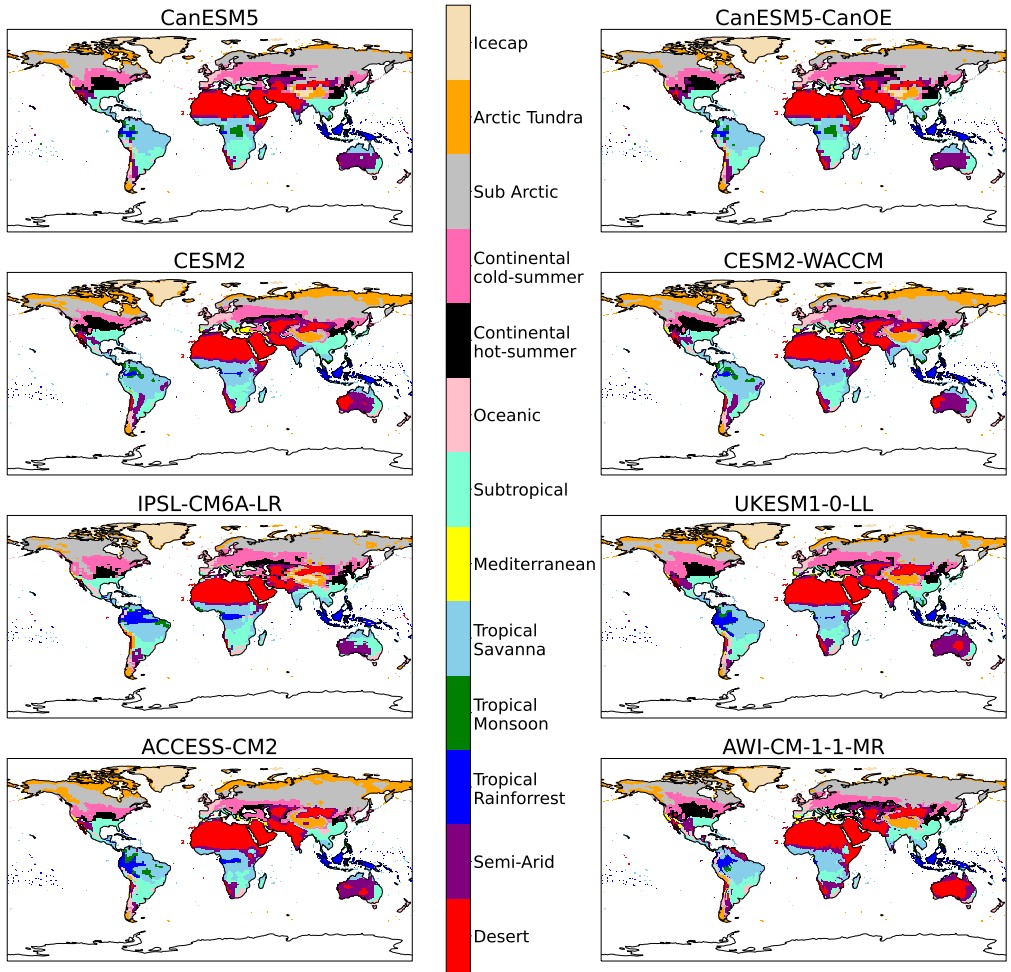

**Figure B1.** Maps of streamlined KG classifications for each model for the reference period (1901 - 1931), without anomaly correction.

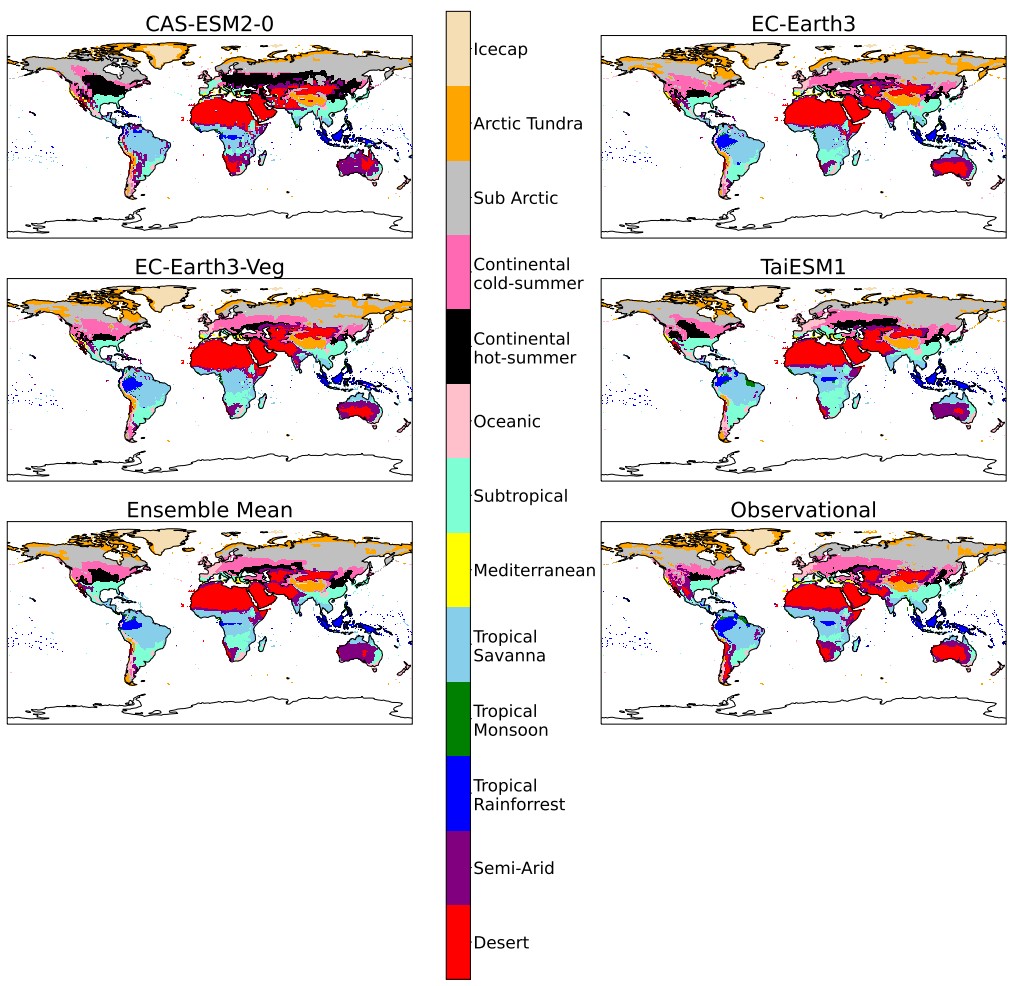

**Figure B1.** Continued.

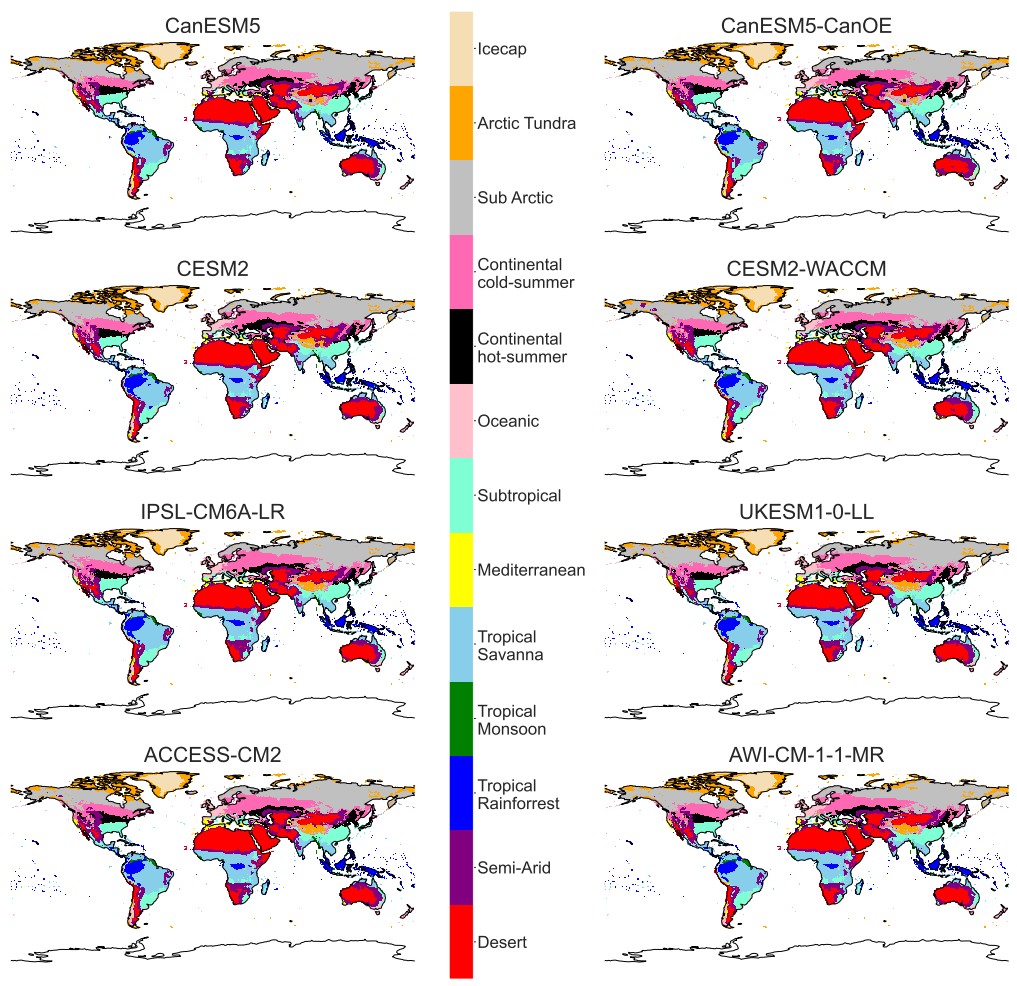

**Figure B2.** Maps of streamlined KG classifications for each model at +1K, with anomaly correction.

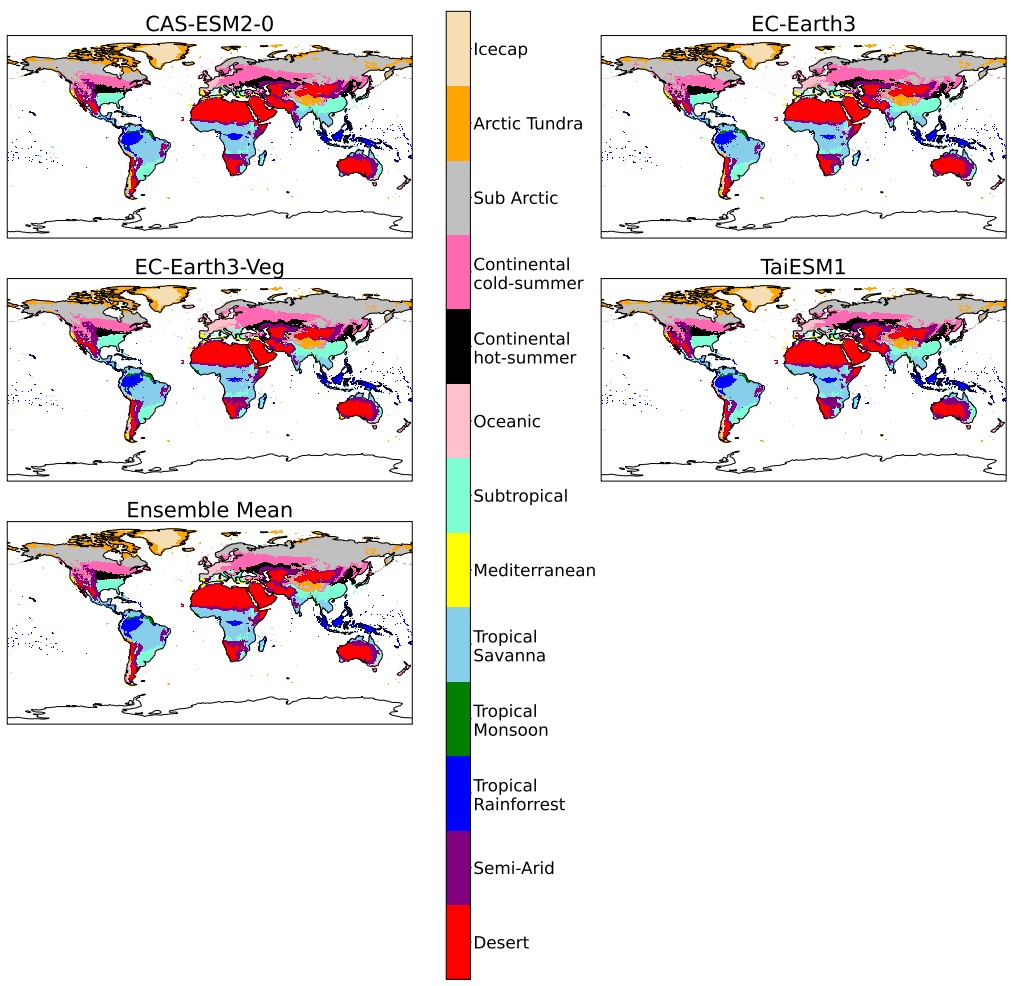

**Figure B2. Continued.**

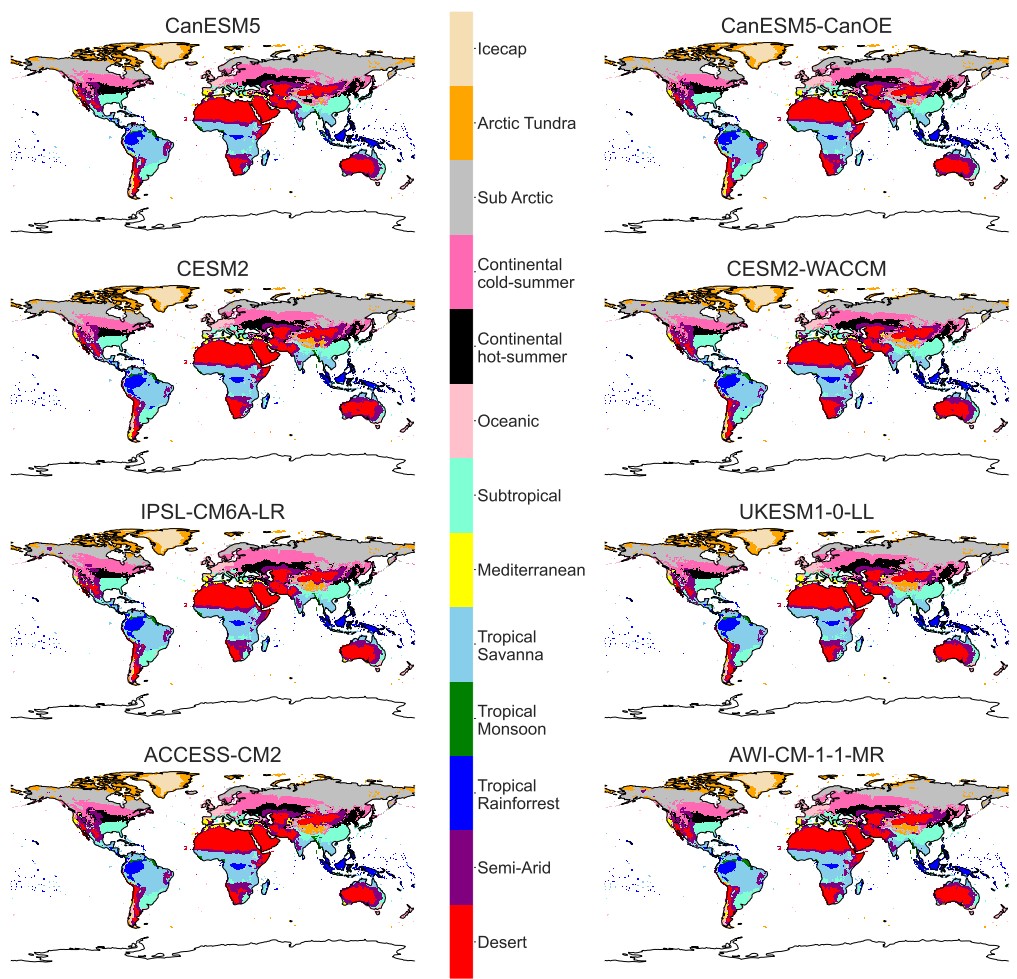

**Figure B3.** Maps of streamlined KG classifications for each model at +1.5K, with anomaly correction.

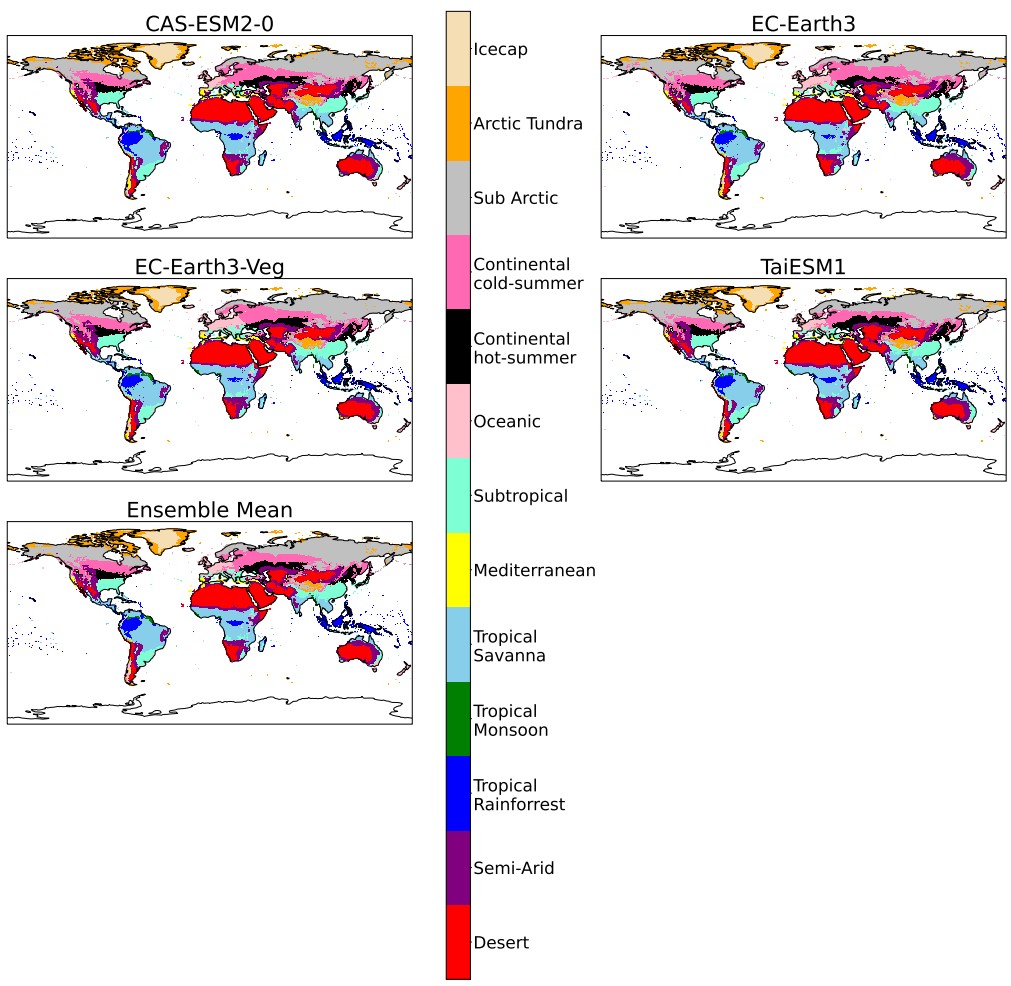

**Figure B3. Continued.**

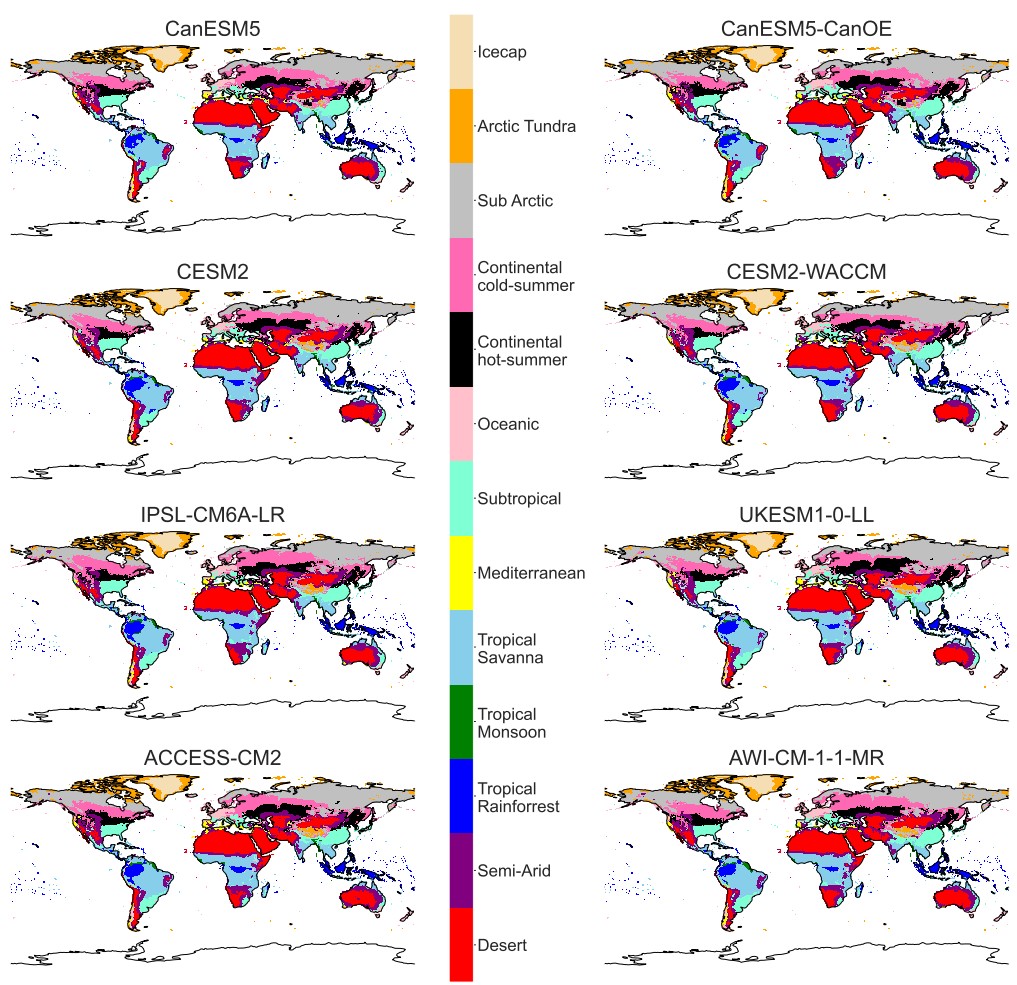

**Figure B4.** Maps of streamlined KG classifications for each model at +2K, with anomaly correction.

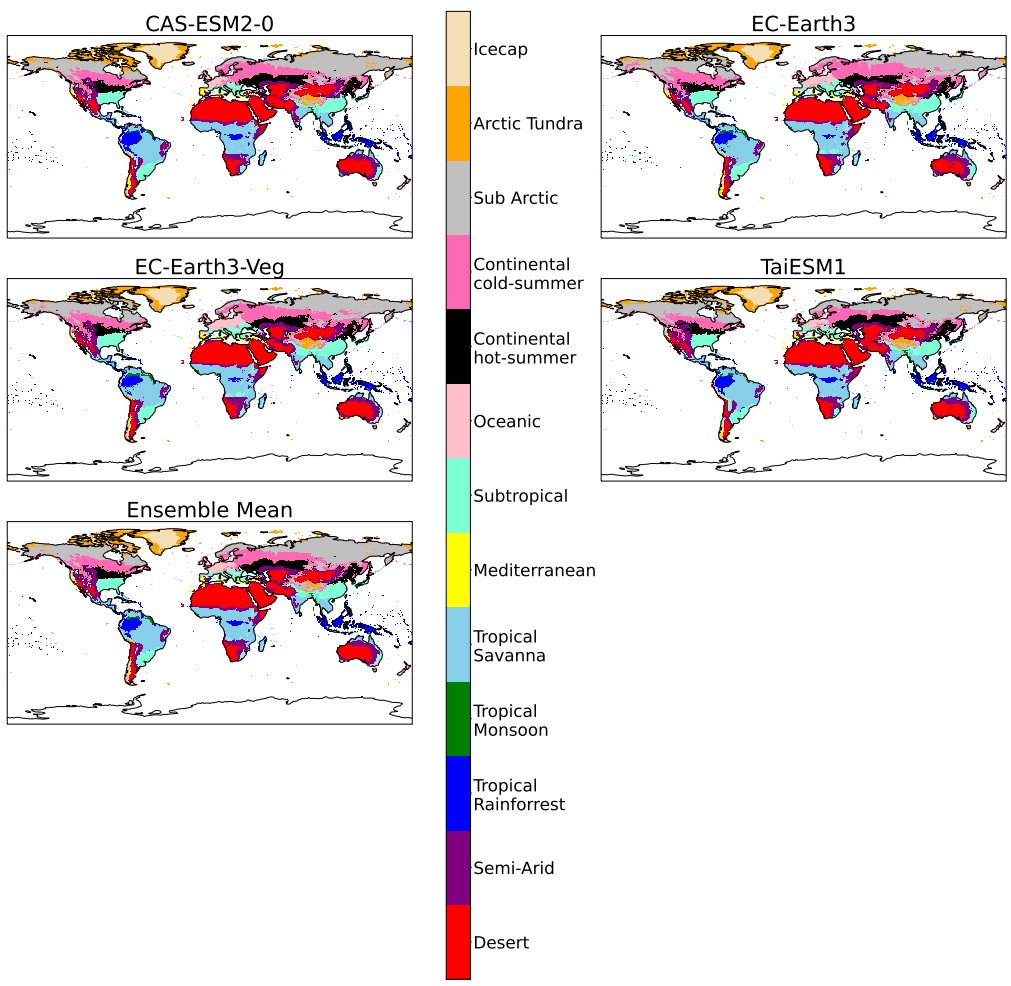

**Figure B4. Continued.**

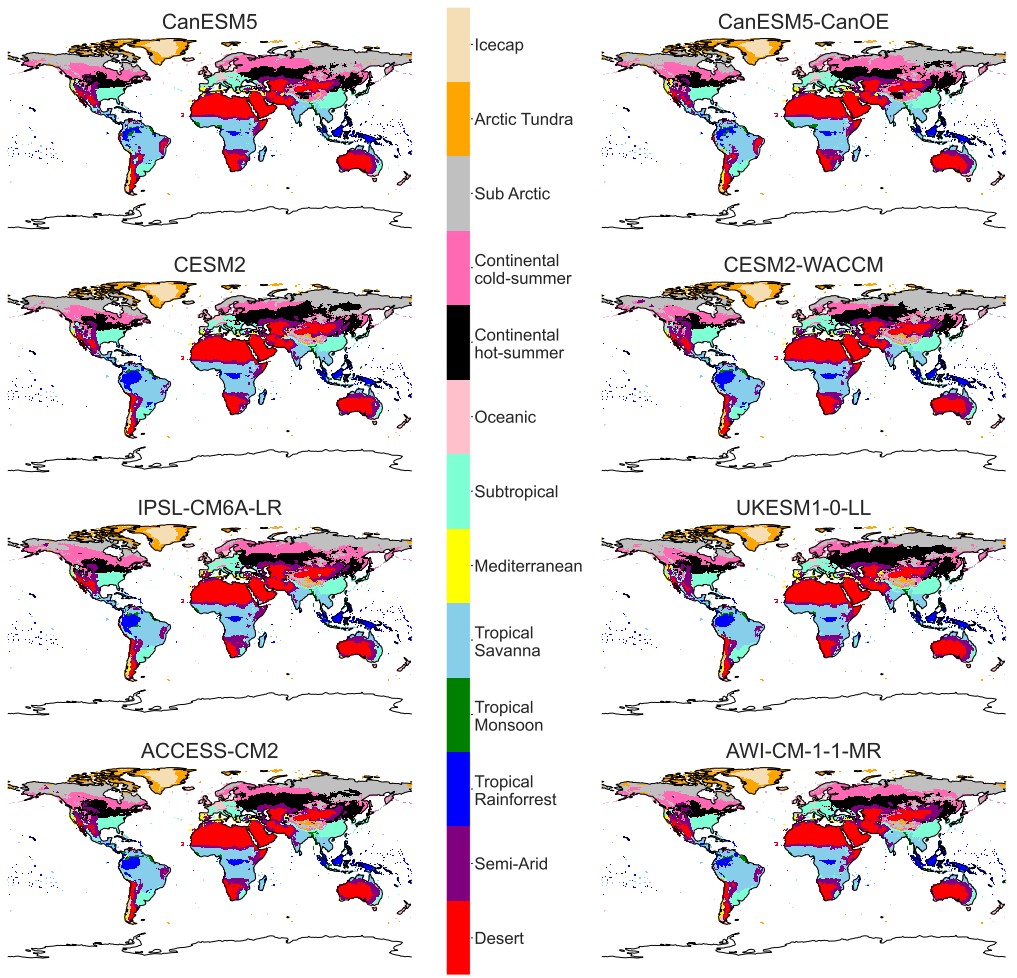

**Figure B5.** Maps of streamlined KG classifications for each model at +3K, with anomaly correction.

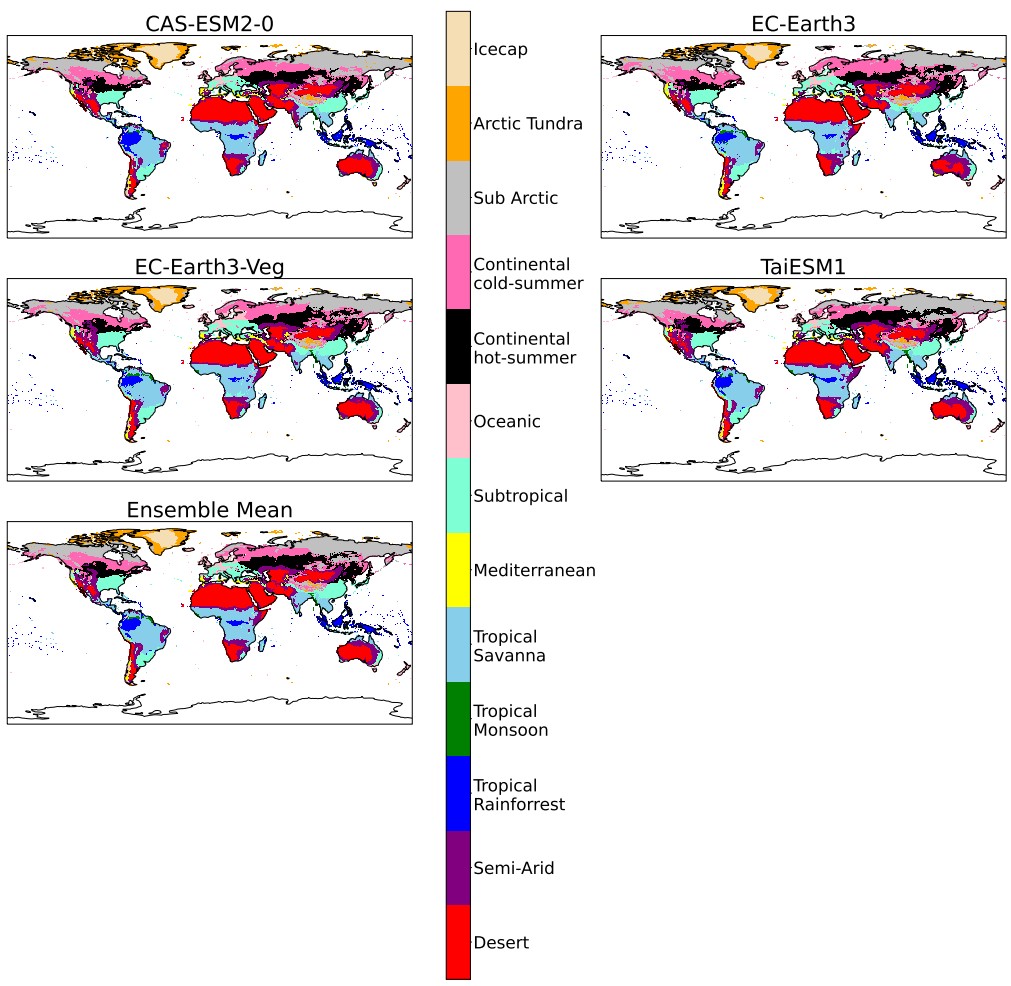

**Figure B5. Continued.**

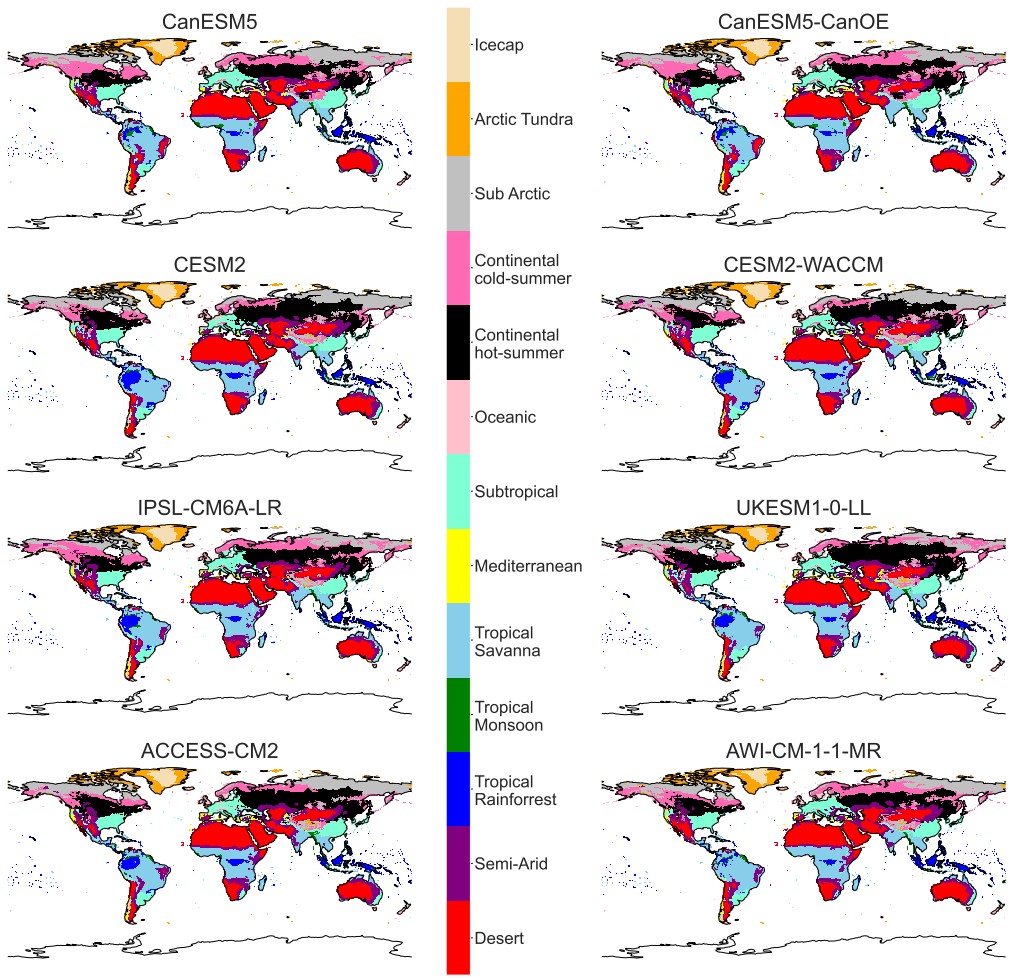

**Figure B6.** Maps of streamlined KG classifications for each model at +4K, with anomaly correction.

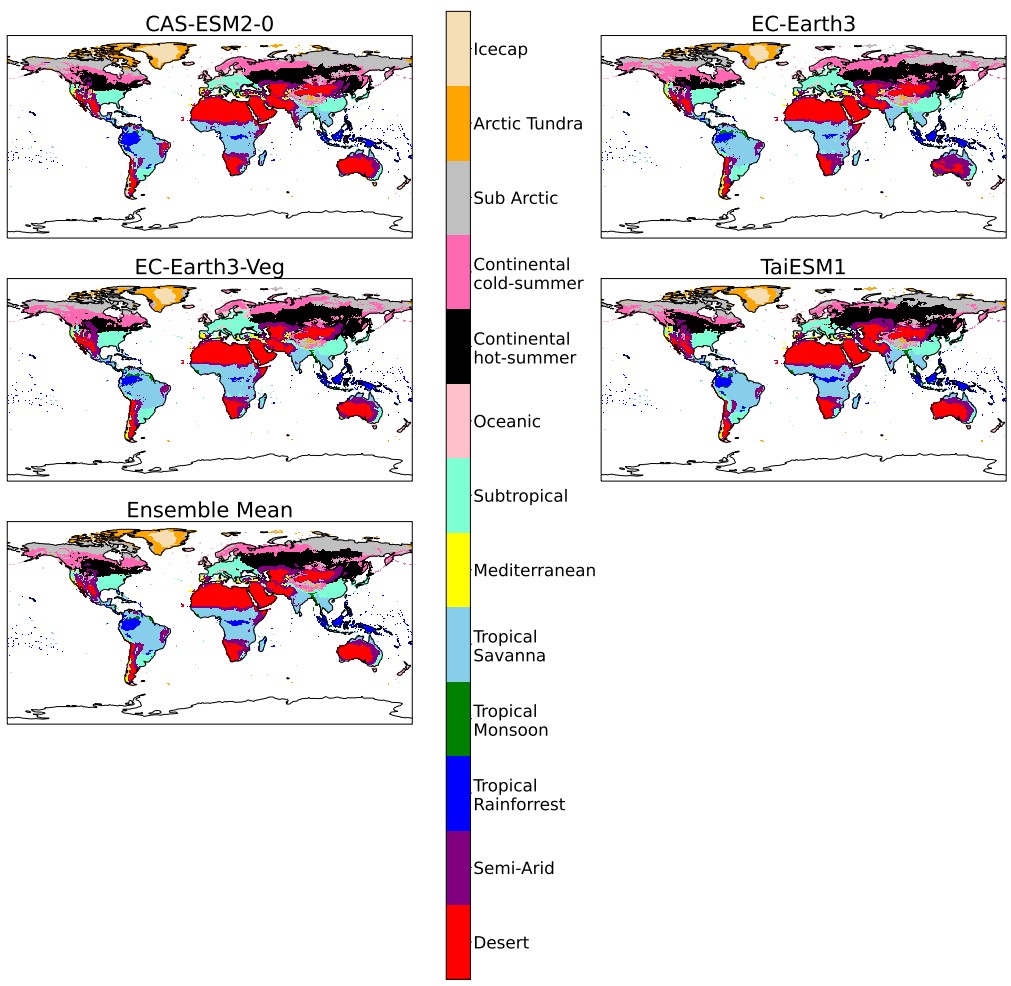

**Figure B6. Continued.** .

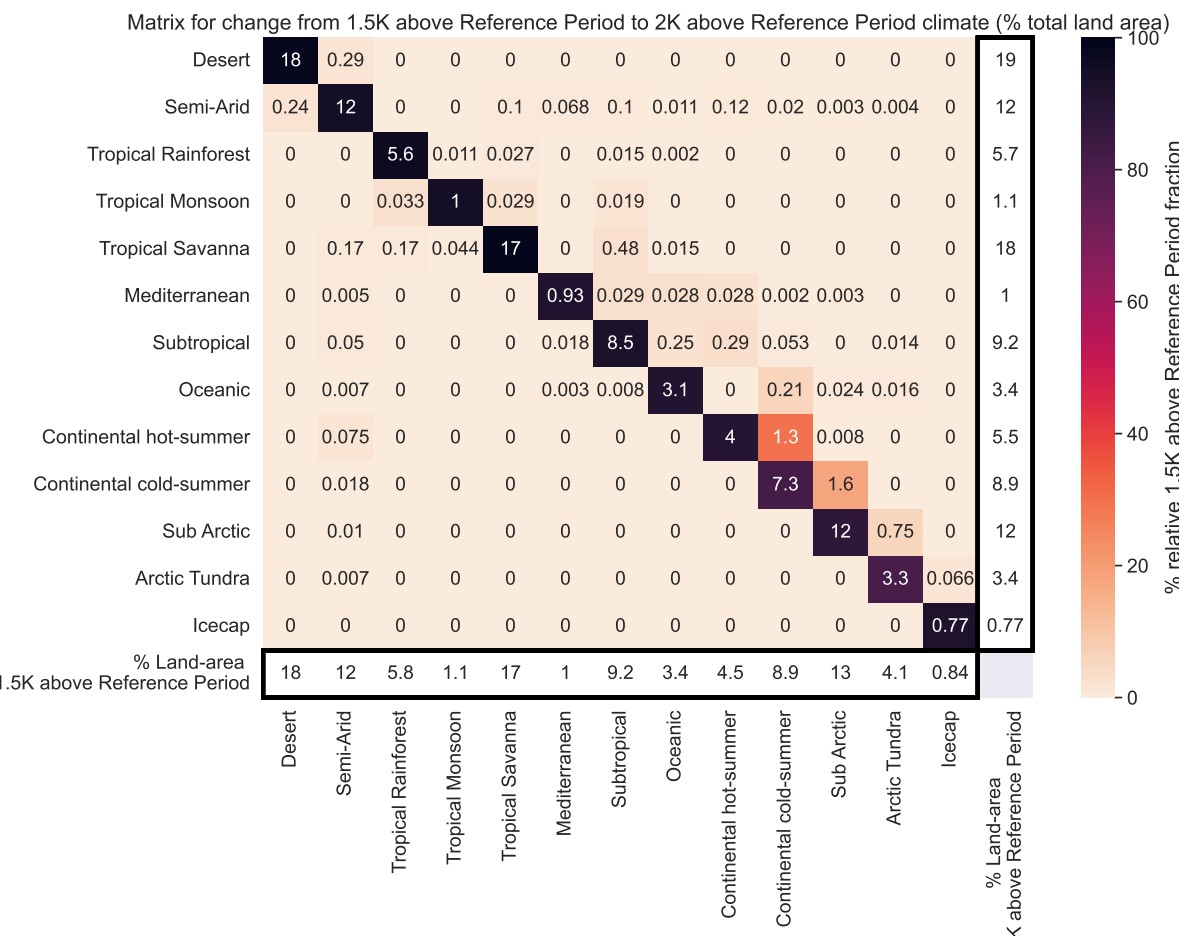

**Figure C1.** Land area bioclimate classification change between 1.5K and 2K of global warming.

## Appendix D

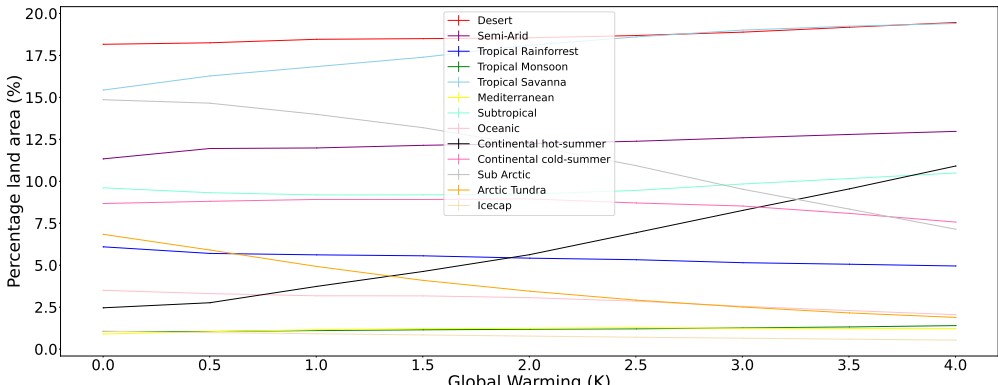

**Figure D1.** Land area distribution of individual streamlined classifications, Classifications that show large growth in their coverage include Continental hot-summer and Tropical Savanna. Classifications that show major reductions include Icecap, Arctic Tundra, and Sub Arctic.

*Author contributions.* MS carried out the data analysis and drafted the paper. MW & PC advised on the study. All authors contributed to the submitted paper.

*Competing interests.* No competing interests are present.

*Acknowledgements.* The authors acknowledge funding from the ERC ECCLES project (PC and MW) and the EU 4C project (PC). We also acknowledge the World Climate Research Programme's Working Group on Coupled Modelling, which is responsible for CMIP, and we thank the climate modelling groups for producing and making available model output from the CMIP6 models.

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
