# Peer review of "Bioclimatic change as a function of global warming from CMIP6 climate projections"

_Biogeosciences, 2022_

## Author Response (AR1)

Biogeosciences Discuss., author comment AC1
https://doi.org/10.5194/bg-2022-74-AC1, 2022
**Reply on RC1**

Morgan Sparey et al.
* * *
Author comment on "Bioclimatic change as a function of global warming from CMIP6 climate projections" by Morgan Sparey et al., Biogeosciences Discuss., https://doi.org/10.5194/bg-2022-74-AC1, 2022
* * *
**We thank the reviewer for their comments. Our responses below are in bold.**

One potential additional improvement to the manuscript would be to develop equations, like that in Equation 1 for the % land area changing climate type, for each streamlined classification climate type of Table 2. As warming increases from 0K to 4K some of the streamlined climate types will increase in % land area covered and others will decrease. An equation for each streamlined climate type could be very interesting / useful as some climate types will expand and reduce at different rates compared to the global land area change in Equation 1. Adding an extra column to Table 2 with the equation for each climate type would be very useful additional information. This would allow researchers interested in particular bioclimates to use these results for their research.

**Many thanks for this suggestion, which is very helpful. We will include the additional analysis/equations, as suggested.**

Specific comments

Line 53: change "data to to" to "data to"

**Done.**

Table 1: while the following differences from Peel et al (2007) are largely minor and most likely do not impact the end results significantly, it is important to note them. The Ds climate correction is likely to be the most important and should be corrected.

- Criteria for C climate: change from "$0ËC ≤ Tmin <18ËC$, $Tmax ≥ 10ËC$" to "$0ËC < Tmin <18ËC$, $Tmax ≥ 10ËC$".
- Criteria for Cs climate: change from "$Pwwet ≥ 3*Psdry$, $Psdry < 4$" to "$Pwwet > 3*Psdry$, $Psdry < 4$".
- Criteria for D climate: change from "$Tmin < 0ËC$, $Tmax ≥ 10ËC$" to "$Tmin ≤ 0ËC$, $Tmax ≥ 10ËC$".
- Criteria for Dw climate: change from "$Pswet ≥ 10*Pwdry$" to "$Pswet > 10*Pwdry$".
- Criteria for Ds climate: change from "$3*Psdry < Pwwet$" to "$3*Psdry < Pwwet$, $Psdry < 4$".
- Criteria for ET climate: change from "$0ËC ≤ Tmax <10ËC$" to "$0ËC < Tmax <10ËC$".

**The correct (Peel et al., 2007) criteria were actually used in our analysis. These issues are all typos in Table 1, which have now been corrected.**

Line 63: change "First, C and D climates follow a 0Ë□C threshold instead of 3Ë□C" to "First, C and D climates follow a 0Ë□C threshold instead of -3Ë□C".

**Done.**

Line 75: I know data can now be considered as singular or plural, but I recommend changing "model and observational data is smoothed" to "model and observational data are smoothed".

**Done.**

Line 85: what do you mean by "anomaly corrected fields"? Not all readers will understand this term or what it means, so more explanation is required.

**By "anomaly corrected field" we mean climate model outputs that have been corrected to agree with the observational record over the overlapping period of 1901-1931. This is done by calculating anomalies relative to that period for each model and then adding these anomalies to the observational climatology. The following text has been added to clarify this: "Model outputs are *anomaly corrected* to agree with the observational over the period 1901-1931. This is done by calculating anomalies relative to that period for each model and then adding these anomalies to the observational climatology."**

Table 2: In Table 1 all second letters were capital (for example CFa rather than Cfa). However, in Table 2 a mixture of second letter capitalisation is used (see Subtropical). Please be consistent.

**As suggested, second letters in Table 2 have now all been capitalised.**

Figure 3: It would be better to increase the size of these four maps as it is very hard to see the differences when the maps are so small. Rather than one column of four maps, try two columns of two maps. Also, why are these KG maps called anomaly plots?

**We find that four larger maps make it more difficult to compare the plots. The page width limitations also mean that two columns of figure 3 show a negligible improvement. Therefore to improve legibility, larger versions of these maps have instead been added to the appendix.**

**For clarity, we now refer to these maps as "anomaly-corrected maps".**

Figure 5a & 5b: the right column of numbers next to the colour bar is labelled "% Land-area 4K" in both 5a and 5b. I think this should be "% Land-area 1.5K" for 5a and "% Land-area 2K" for 5b.

**Indeed, that you for finding this typo, which has now been corrected.**

Line 142: You refer to Figure 5a, but don't you mean Figure 5c? Figure 5a shows the 1.5K results, whereas Figure 5c shows the 4K results. Hence the comment about Arctic Tundra should be updated to 75% less land-area.

**Thanks again. This has now been fixed to refer to Figure 5c.**

Equation 1: you provide an equation, but no measure of how well this model fits the data. I realise there are only nine data points supporting this model fit, but a metric like $R^2$ would be useful to indicate how well the model fits the data.

**As suggested, the $r^2$ values of the fits are now included.**

[Figure]

Biogeosciences Discuss., author comment AC2
https://doi.org/10.5194/bg-2022-74-AC2, 2022
**Reply on RC2**

Morgan Sparey et al.
* * *
Author comment on "Bioclimatic change as a function of global warming from CMIP6 climate projections" by Morgan Sparey et al., Biogeosciences Discuss., https://doi.org/10.5194/bg-2022-74-AC2, 2022
* * *
**We thank the reviewer for their comments. Our responses below are in bold.**

**Specific comments**

I think the most critical point the authors should address is the lack of a scientific "discussion". There is no single reference to other works in Section 3. For instance, how do these results compare with Kim and Bae (2021)? Are these results far away from similar works with CMIP5 like Rahimi et al. (2020)? Here I only refer to those citations in the Introduction, but perhaps there exists more literature that can be discussed. Keeping together "Results and Discussion" is possible, but in this case I think Section 3 only includes a description of the outcome of the author's own analyses.

**As requested, a longer Discussion section is included which refers to a these additional papers and others.**

Also, besides using references for the discussion section, I think the authors should revise the use of references throughout the manuscript. For instance in the Introduction there are no references about CMIP6 or the Paris climate targets. Also when the authors say in line 60 that the scheme has had many alterations, they could provide a list of some publications in parentheses with as e.g..

**We have included these suggested references and expanded our reference list for the paper.**

 Is Table 1 and the modifications described in lines 62-65 the same as in Peel et al. (2007)? Or are these the author's own modifications to what Peel et al. (2007) do? Please also consider including a citation in the caption of Table 1 if this is indeed taken from the reference.

**These classifications follow Peel et al. (2007) exactly. Although this paper is cited directly prior to the table, it will also now be referenced in the table caption.**

Authors should motivate better their model selection process. This is especially important since the 6 chosen models include repeated model components. How different are, for instance, CanESM5 and CanESM5-CanOE in terms of simulated (atmospheric variables)

monthly precipitation and temperature values? What are the "data management" reasons that lead to these 6 models being selected over other models?

**These models were chosen based on the requirements for our analysis. Models needed to provide a "land" (sftlf) mask, and to output the required fields (monthly-mean temperature and precipitation) over each year of the 250 year runs, in a single file.**

Along with the previous comment, please include a table with the specifications of the data used. Since the model output data is a critical component of this study, it is important that some characteristics (e.g., spatial resolution, time step, citation to techical paper) can be readily seen and compared in a table.

**A table with these key characteristics of each model will be included in the revised manuscript.**

Please discuss possible shortcomings of downscaling model output to a finer grid (0.5°), in case this was done when the model output is at a coarser scale. This is why a table with some specifications of the data could be useful.

**We now include the following text: "The model output data is typically at a coarser resolution than the underlying 0.5° climatology. The anomaly corrected fields therefore contain spatial variability that is solely due to the underlying climatology at scales which are not resolved by a model. This also implies that the diagnosed changes in bioclimatic types (which are dependent on the model anomalies) tend to be somewhat smoother at these finer spatial scales".**

Please consider providing maps like those in Appendix A but using the "streamlined" version.

**We include maps for the streamlined classification in the Appendix, as suggested.**

In multiple occasions the word "significant" is used. I think that with the data the authors have it should be possible to perform some statistical significance tests. I think such tests could increase the strength of the results. Without the tests, please consider alternating with synonyms like "substantial", "marked", "large" or alike.

**As requested, "significant" is only used to refer to statistical significance in the revised paper. The suggested synonyms are used elsewhere.**

Technical details

Please consider re-organising some paragraphs in the Introduction. Paragraph starting on line 35 may fit better after the description of the classification systems. As a side note, this paragraph does not mention any references about CMIP6.

**We have revised this paragraph to include references to CMIP6, as suggested.**

I would recommend to move the sentence about previous applications of KG (line 43) to the end of Section 2.1.

**We agree, this change has been implemented.**

Abstract contains references. Are they urgently required?

**Good point. These references have been removed from the abstract.**

Consider prefixing "ensemble mean" with "multimodel", because some times "ensemble" could be referring to a group of runs or data.

**We agree that this change improves clarity and we have therefore implemented it as suggested.**

Please include spaces between numbers and their units, and use units in exponential notation (e.g., cm month-1)

**Done.**

I think authors should mention vegetation in some places. At least when explaining the KG classification system on line 32.

**As suggested, we have expanded on the description of the KG classification system to give examples of the vegetation types that it suggests for particular climates.**

Perhaps Section 2.4.1 should actually be a subsection of or follow the Section on the traditional KG scheme (Section 2.1).

**Done. We agree that following the traditional scheme is a more logical position for the streamlined scheme, so this change has been made.**

Please expand on what the anomaly corrections are.

**The following text has been added to clarify this: "Model outputs are *anomaly corrected* to agree with the observational over the period 1901-1931. This is done by calculating anomalies relative to that period for each model and then adding these anomalies to the observational climatology.**

Consider using "averaging" instead of "meaning".

**We prefer to continue to use "meaning" as it more accurately describes the averaging process used.**

 L3: "hinder" -> maybe "limit" is an alternative. Many detailed assessments can be made in spite of inter-model spread.

**As suggested, "hinder" has been replaced with "limit".**

L4: why capitalisation in "Earth System Models".

**Earth System Models is now only capitalised in its first use, subsequently it has been replaced with ESM.**

L4: remove "very".

**Done.**

L5: "will" -> "would".

**Done.**

L21: Is it correct "regional areas"?

**Changed to "areas".**

L31: missing umlaut in Köppen name.

**Corrected.**

L43: maybe this "most popular" should be referenced. Otherwise "popular" should suffice.

**Changed to "highly popular".**

review "more intuitive".

**Changed to "intuitive".**

L52: review this sentence.

**Changed from:**

**"Furthermore, we utilise the KG system as an exploratory technique for understanding CMIP6 model output, the KG classification scheme has been previously applied to CMIP5 data to to evaluate simulations (Phillips and Bonfils, 2015)"**

**to:**

**"We utilise the KG system as an exploratory technique to visualize CMIP6 model output. The KG classification scheme has been previously applied to CMIP5 data to evaluate simulations (Phillips and Bonfils, 2015)"**

L60: check citation is \textcite, not \parencite.

**Fixed.**

L72: please use active voice: "we do not expect".

**Done.**

L76: check if "correctly" can be replaced by "following observations".

**Changed to "of observational data".**

L80: is it above pre-industrial levels or above the reference period 1901--1931?

**Changed to reference period.**

- L81: remove "model" after "CMIP6".

**Done.**

- L90: review this sentence.

**Changed from**

**"A key goal of bioclimatic classifications is to illustrate climate change in a way that is more intuitive for many people."**

**To**

**"A key goal of bioclimatic classifications is to illustrate climate change in a way that is intuitive."**

- L110: remove period after "Figure 1".

**Done.**

- L116: (and Fig. 1 and 2 captions) consider "averaging process" instead of "meaning".

**See above. We prefer "meaning" as it is less ambiguous.**

- L117: 'lagging' as in having lower values?

**"Lagging" as in following the same trend as the individual models but behind them in time and with reduced variability, this has been clarified in the revised manuscript.**

- L119, L120: check use of "correctly", that can always be related to "according to observation-based KG".

**Now refers to "observational ".**

- L129: check "real world climate".

**Changed to "observed climate data".**

- L134: use cross-referencing with the appendices (e.g., Appendix A).

**Done.**

- L135: "shows" -> "suggests". "will be" -> "could be".

**As suggested, these changes have been made.**

- Fig. 3 caption: maybe place "reference period" before CMIP6 ssp585 runs.
**A semicolon has been added for clarity:**

**"Anomaly plot of ensemble mean CMIP6 ssp585 runs for; the reference period, 1.5K, 2K, and 4K of global warming with the traditional Köppen-Geiger classification system applied."**

- L162: "will" -> "could".

**Done.**

---

## Author Response (AR2)

We thank the reviewers for their comments which have improved our manuscript again. In particular, report 2's suggestion of adding uncertainty quantification into our results. Agreement between the models is surprisingly high even at 4K of warming. This gives our results a pleasing degree of robustness. Our responses below are in bold.

**Response to report 1.**

The authors present a thorough analysis of model output from climate models participating in CMIP6. They show useful maps, metrics and equations that inform potential global and regional bioclimatic change due to warming. The manuscript is clearly outlined, well written and fits the journal scope.

However, as a minor comment, I think the authors should expand their discussion of the results by addressing perhaps two topics: (a) evidence of change: what evidence/examples/literature exists of warming-induced bioclimatic change? Authors mention desertification in the Amazon, as well as changes in North America and Northern Eurasia. What does the literature say about bioclimatic change in these regions?

**We agree that further discussion of the evidence of change would yield useful information around the context of the projections. We have included discussion about observed biome changes in the boreal forest and the Amazon, along with literature confirming these changes. The following text has been added to section 3.2 lines 174-180: "Although the KG scheme exclusively maps climate, the biologic implications of these changes can be seen. The Sub-arctic region has historically been dominated by the boreal forest (Kayes and Mallik, 2020) though there is evidence suggesting a slow northward migration of non-native warm-adapted tree species into historically boreal areas (Viacheslav et al., 2007; Boisvert-Marsh et al., 2014). However, the rate of global warming far exceeds the rate of boreal migration due to limits on the tree's ability to migrate (McKenney et al., 2007). With the continued reduction of the sub-arctic bioclimate we predict a likely continuation of these trends. The indicated shift in the Amazon from tropical rainforest to savanna can also be seen, an expansion of white-sand savannas in the Amazon has already been found (Flo, 2021)."**

And (b) how changes happen: how does warming induce the bioclimatic change? How do regions shift from "colder-wetter" to "warmer-drier"? can authors speculate on a timescale of change? Maybe some of the changes in Figure 4 would happen faster than others?

**The following text has been added to section 2.1 lines 71-80: "The outcome of competition between different plant types varies depending on the climatic conditions, such that the long-term equilibrium biome (which is a mixture of different plant types) will also vary with the climate. This is the underlying basis for bioclimatic schemes such as Koppen-Geiger, which is used here to classify the climate rather than to predict biome shifts. Changes in the actually distribution of vegetation also depend on the direct effects of changes in carbon dioxide and nitrogen availability, and may take decades to materialise (e.g. because the rate of climate change is significant compared to the characteristic multi-decadal timescales of a forest). These changes are best predicted with complex Dynamical Global Vegetation Models (DGVMs), which are based-on detailed representations of plant physiology and demographic processes (Argles, 2022). Although bioclimatic schemes are no substitute for DGVMs to predict vegetation changes, they have the advantage of being transparently simple and offering a more intuitive demonstration of the nature of a projected climate change. It is in this spirit that the Koppen-Geiger bioclimatic scheme is applied in this study."**

These are only some questions that I think could be discussed by the authors. Perhaps another question is: where are the 7.5M km2 that would escape major change?

**We have answered this in the discussion with the following sentences, lines 205-208:**

*'…. Bioclimatic change for over 7.5 million square kilometres of land. These bioclimatic changes being increases of 1% in land area of desert, tropical savanna and continental hot summer biomes. Sub-Arctic and Arctic tundra decrease by 1 and 0.7% respectively. Between 1.5K and 2K there are relatively small or no changes between the remaining classifications in figure 5a and figure 5b. '*

In summary, I think the manuscript would benefit from additional remarks in the discussion section, linking always the global and regional scales of the bioclimatic changes, and also looking, if possible, at literature outside of the modelling approach.

**We have now included literature for evidence of change outside of modelling as outlined previously in lines 174-180.**

Technical details.

L9: drier

**Fixed**

L71: perhaps bring out as a subsection (i.e. as 2.2 instead of 2.1.1)

**We feel that as the streamlined scheme is made entirely from the traditional, it is clearer when presented as a subsection of the traditional scheme.**

L87: Table 3

**Fixed**

L157: Please review. Maybe I am reading the matrix wrong but in Fig. 5c, Arctic Tundra goes from 6.8 to 1.8. That is a reduction of over 70%

**This has been reviewed and is correct.**

**Response to report 2.**

Analyzing Earth System models to explore regions that will undergo major bioclimatic transitions is certainly of general interest nowadays. I found this paper scientifically sound and well written. As a new reviewer, I only have two more comments for the authors.

Have the authors explored the ensemble median rather than the mean? This may be more trustworthy than the mean as it is less impacted by an outlier (e.g., if a specific model has a very different prediction from the others). I suggest exploring this option.

**As usual in climate modelling, we choose to focus primarily on the ensemble mean, although we note that ensemble mean and ensemble median climates have been found to be very similar. We have now included acknowledgement of, and justification for this decision. This can be found in section 2.4 lines 122-124: "As usual in climate modelling, we focus primarily on the ensemble**

**mean, although we note that ensemble mean and ensemble median climates have been found to be very similar (Sillmann et al., 2013; Kim et al., 2020; Li et al., 2021)."**

Also, when analyzing these models, uncertainty quantification is very important, and I was surprised that here uncertainty was not explored much. One can create a map with shifts in bioclimate classification, but the meaning of this projected shift is different in each grid cell because the uncertainty (intermodal variability) is different. I think this could be addressed by adding (at least in appendix) maps of the uncertainty (e.g., standard deviation) for 1.5, 2, and 4K. This would allow someone interested in these bioclimatic shifts to see the confidence of these projections.

**We have now included an indication for the agreement between the models on a particular classification or classification change. Hatching has been implemented in figures 3 and 4 to indicate areas where there is less than 66% agreement between the models. This demonstrates the confidence in the results given in each cell grid.**